# Conditional Generative Modeling for High-dimensional Marked Temporal Point Processes

## Abstract

Point processes offer a versatile framework for sequential event modeling. However, the computational challenges and constrained representational power of the existing point process models have impeded their potential for wider applications. This limitation becomes especially pronounced when dealing with event data that is associated with multi-dimensional or high-dimensional marks such as texts or images. To address this challenge, this study proposes a novel event generative framework for modeling point processes with high-dimensional marks. We aim to capture the distribution of events without explicitly specifying the conditional intensity or probability density function. Instead, we use a conditional generator that takes the history of events as input and generates the high-quality subsequent event that is likely to occur given the prior observations. The proposed framework offers a host of benefits, including considerable representational power to capture intricate dynamics in multi- or even high-dimensional event space, as well as exceptional efficiency in learning the model and generating samples. Our numerical results demonstrate superior performance compared to other state-of-the-art baselines.

## 1 Introduction

Point processes are widely used to model asynchronous event data ubiquitously seen in real-world scenarios, such as earthquakes (Ogata, 1998; Zhu et al., 2021a), healthcare records (Dong et al., 2023), and criminal activities (Mohler et al., 2011). These data typically consist of a sequence of events that denote when and where each event occurred, along with additional descriptive information such as category, locations, and even text or image, commonly referred to as "marks". With the rise of complex systems, advanced models that go beyond the classic parametric point processes (Hawkes, 1971) are craved to capture intricate dynamics involved in the data generating mechanism.

Neural point processes (Shchur et al., 2021), such as Recurrent Marked Temporal Point Processes (Du et al., 2016) and Neural Hawkes (Mei & Eisner, 2017), are powerful methods for event modeling and prediction. They use neural networks (NNs) to model the intensity of events and capture complex dependencies among observed events. However, due to the use of NNs, the cumulative (*i.e.*, integral of) conditional intensity in the point process likelihood is often analytically intractable, requiring complex and expensive approximations during learning. More seriously, these models face significant limitations in *generating events with high-dimensional marked information*, as the event simulation relies heavily on the thinning algorithm (Ogata, 1981), which can be costly or even impossible when the mark space is high-dimensional. This significantly restricts the applicability of these models to modern applications (Zhu & Xie, 2022; Williams et al., 2020), where event data often come with high-dimensional marks, such as texts and images in police crime reports or social media posts.

Recent developments in generative modeling (Kingma & Welling, 2014; Sohn et al., 2015; Ho et al., 2020) offer potential solutions to handling high-dimensional event marks in point processes. These generative models are adept at producing high-quality content based on contextual cues. Their effectiveness is rooted in their ability to approximate the underlying high-dimensional data distribution through generated samples, rather than directly estimating the density function.

This paper presents a novel generative framework for modeling point processes with high-dimensional marks. Unlike traditional point process models that depend on defining the conditional intensity or probability density function through parametric models (Du et al., 2016; Mei & Eisner, 2017;

Shchur et al., 2021; Dong et al., 2022), our model estimates the distribution of events using samples generated by a conditional generator, which takes the history of events as its input. The event history is summarized by a recurrent neural architecture, allowing for flexible selection based on the application's needs. The benefits of our model can be summarized by:

1. Our model is capable of handling high-dimensional marks such as images or texts, an area not extensively delved into in prior marked point process research;
2. Our model possesses superior representative power, as it does not confine the conditional intensity or probability density of the point process to any specific parametric form;
3. Our model outperforms existing state-of-the-art baselines in terms of estimation accuracy and generating high-quality event series;
4. Our model excels in computational efficiency during both the training phase and the event generation process. In particular, our method needs only $\mathcal{O}(N_T)$ for generating $N_T$ events, in contrast to the thinning algorithm's complexity of $\mathcal{O}(N^d \cdot N_T)$, where $N \gg N_T$ and $d$ represents the event dimension.

It is important to note that our proposed framework is general and model-agnostic, meaning that a wide spectrum of generative models and learning algorithms can be applied within our framework. In this paper, we present two possible learning algorithms and evaluate our framework through extensive numerical experiments on both synthetic and real-world data sets.

**Related work**   Seminal works in point processes (Hawkes, 1971; Ogata, 1998) introduce self-exciting conditional intensity functions with parametric kernels. While these models have proven useful, they possess limitations in capturing the intricate patterns observed in real-world applications. Recently, there has been a focus on enhancing the expressiveness of point process models through the integration of neural networks. Landmark studies like recurrent marked temporal point processes (RMTPP) (Du et al., 2016) and Neural Hawkes (Mei & Eisner, 2017) have leveraged recurrent neural networks (RNNs) to embed event history into a hidden state, which then represents the conditional intensity function. Another line of research opts for a more "parametric" way, which only replaces the parametric kernel in the conditional intensity using neural networks (Dong et al., 2023; Okawa et al., 2021; Zhu et al., 2021a; 2023). However, these methods can be computationally inefficient, and even intractable, when dealing with multi-dimensional marks due to the need for numerical integration. To overcome these computational challenges, alternative approaches have been proposed by other studies (Chen et al., 2020; Omi et al., 2019; Shchur et al., 2020; Zhou et al., 2022). These approaches focus on modeling the cumulative hazard function or conditional probability rather than the conditional intensity, thereby eliminating the need for integral calculations. Nonetheless, these methods are typically used for low-dimensional event data, and still rely on the thinning algorithm for simulating or generating event series, which significantly limits their applicability in modern applications. Some studies (Dong et al., 2022; Zhu & Xie, 2022) propose to model high-dimensional marks by considering simplified finite mark space. However, work on handling high-dimensional marks in point processes is still limited.

Our research paper is closely connected to the field of generative modeling, which aims to generate high-quality samples from learned data distributions. Prominent examples of generative models include generative adversarial networks (GANs) (Goodfellow et al., 2014), variational autoencoders (VAEs) (Kingma & Welling, 2014), and diffusion models (Ho et al., 2020; Sohl-Dickstein et al., 2015b; Song et al., 2020). Recent studies have introduced conditional generative models (Mirza & Osindero, 2014; Sohn et al., 2015) that can generate diverse and well-structured outputs based on specific input information. In our work, we adopt a similar technique where we consider the history of observed events as contextual information to generate high-quality future events. Furthermore, more recent advancements in this field (Ajay et al., 2022; Li et al., 2020) have extended the application of conditional generative models to a broader range of settings. For example, Ajay et al. (Ajay et al., 2022) uses a conditional generator to explore the optimal decision that maximizes the reward, drawing inspiration from reinforcement learning.

The application of generative models to point processes has received limited attention in the literature. There are three influential papers (Li et al., 2018; Sharma et al., 2019; Xiao et al., 2018) that have made significant contributions in this area by using RNN-like models to generate future events. While RNNs are commonly used for prediction, these papers assume the model's output follows a Gaussian distribution, enabling the exploration of the event space, albeit at the cost of limiting the representational power of the models. To learn the model, they choose to minimize the "similarity"

(*e.g.*, Maximum Mean Discrepancy or Wasserstein distance) between the generated and the observed event sequences. It is important to note that these metrics are designed to measure the discrepancy between two distributions in which each data point is assumed to be independent of the others. This approach may not always be applicable to temporal point processes, particularly when the occurrence of future events depends on the historical context. A similar concept of modeling point processes using conditional generative models is also explored in another concurrent paper (Lin et al., 2022). However, their approach differs from ours in terms of the specific architecture used. They propose a diffusion model with an attention-based encoder, while our framework remains model-agnostic, allowing for greater flexibility in selecting different models. Additionally, their work primarily focuses on one-dimensional events and does not account for multi- or high-dimensional marks.

## 2 METHODOLOGY

### 2.1 BACKGROUND: MARKED TEMPORAL POINT PROCESSES

Marked temporal point processes (MTPPs) (Reinhart, 2018) consist of a sequence of *discrete events* over time. Each event is associated with a (possibly multi-dimensional) *mark* that contains detailed information of the event, such as location, nodal information (if the observations are over networks, such as sensor or social networks), and contextual information (such as token, image, and text descriptions). Let $T > 0$ be a fixed time-horizon, and $\mathcal{M} \subseteq \mathbb{R}^d$ be the space of marks. We denote the space of observation as $\mathcal{X} = [0, T) \times \mathcal{M}$ and a data point in the discrete event sequence as

$$x = (t, m), \quad t \in [0, T), \quad m \in \mathcal{M},$$

where $t$ is the event time and $m$ represents the mark. Let $N_t$ be the number of events up to time $t < T$ (which is random), and $\mathcal{H}_t := \{x_1, x_2, \ldots, x_{N_t}\}$ denote historical events. Let $\mathbb{N}$ be the counting measure on $\mathcal{X}$, *i.e.*, for any measurable $S \subseteq \mathcal{X}$, $\mathbb{N}(S) = |\mathcal{H}_t \cap S|$. For any function $\phi : \mathcal{X} \rightarrow \mathbb{R}$, the integral with respect to the counting measure is defined as

$$\int_S \phi(x)d\mathbb{N}(x) = \sum_{x_i \in \mathcal{H}_T \cap S} \phi(x_i).$$

The events' distribution in MTPPs can be characterized via the conditional intensity function $\lambda$, which is defined to be the occurrence rate of events in the marked temporal space $\mathcal{X}$ given the events' history $\mathcal{H}_{t(x)}$, *i.e.*,

$$\lambda(x|\mathcal{H}_{t(x)}) = \mathbb{E}\left(d\mathbb{N}(x)|\mathcal{H}_{t(x)}\right)/dx, \tag{1}$$

where $t(x)$ extracts the occurrence time of the possible event $x$. Given the conditional intensity function $\lambda$, the corresponding conditional probability density function (PDF) can be written as

$$f(x|\mathcal{H}_{t(x)}) = \lambda(x|\mathcal{H}_{t(x)}) \cdot \exp\left(-\int_{[t_n, t(x)) \times \mathcal{M}} \lambda(u|\mathcal{H}_{t(u)})du\right). \tag{2}$$

where $t_n$ denotes the time of the most recent event that occurred before time $t(x)$.

The point process models can be learned using maximum likelihood estimation (MLE). The log-likelihood of observing a sequence with $N_T$ events can therefore be obtained by

$$\ell(x_1, \ldots, x_{N_T}) = \int_{\mathcal{X}} \log \lambda(x|\mathcal{H}_{t(x)})d\mathbb{N}(x) - \int_{\mathcal{X}} \lambda(x|\mathcal{H}_{t(x)})dx. \tag{3}$$

See all the derivations in Appendix A.

### 2.2 CONDITIONAL EVENT GENERATOR

The main idea of the proposed framework is to use a *conditional event generator* to produce the $i$-th event $x_i = (t_{i-1} + \Delta t_i, m_i)$ given its previous $i - 1$ events. Here, $\Delta t_i$ and $m_i$ indicate the time interval between the $i$-th event and its preceding event and the mark of the $i$-th event, respectively. Formally, this is achieved by a generator function:

$$g(z, \boldsymbol{h}_{i-1}) : \mathbb{R}^{r+p} \rightarrow (0, +\infty) \times \mathcal{M}, \tag{4}$$

which takes an input in the form of a random noise vector ($z \in \mathbb{R}^r \sim \mathcal{N}(0, I)$) and a hidden embedding ($\boldsymbol{h}_{i-1} \in \mathbb{R}^p$) that summarizes the history information up to and excluding the $i$-th event, namely, $\mathcal{H}_{t_i} = \{x_1, \ldots, x_{i-1}\}$. The output of the generator is the concatenation of the time interval

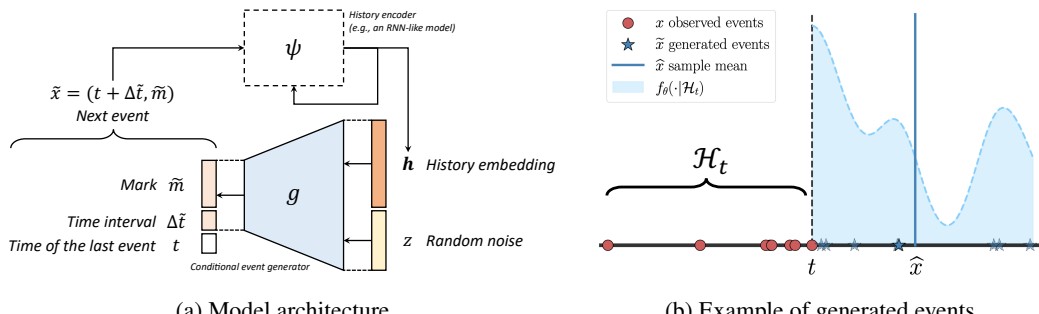

(a) Model architecture            (b) Example of generated events

Figure 1: (a) The architecture of the proposed framework, which consists of two key components: A conditional generative model $g$ that generates $(\Delta\widetilde{t}, \widetilde{m})$ given its history embedding and an RNN-like model $\psi$ that summarizes the events in the history. (b) An example of generated one-dimensional (time only) events $\{\widetilde{x}^{(j)}\}$ given the history $\mathcal{H}_t$. The shaded area suggests the underlying conditional probability density captured by the model parameters $\theta$.

and mark of the $i$-th event denoted by $\Delta\widetilde{t}_i$ and $\widetilde{m}_i$, respectively. To ensure that the time interval is positive, we restrict $\Delta\widetilde{t}_i$ to be greater than zero.

To represent the conditioning variable $\boldsymbol{h}_{i-1}$, we use a *history encoder* represented by $\psi$, which has a recursive structure such as recurrent neural networks (RNNs) (Yu et al., 2019) or Transformers (Vaswani et al., 2017). In our numerical results, we opt for long short-term memory (LSTM) (Graves & Graves, 2012), which takes the current event $x_i$ and the preceding hidden embedding $\boldsymbol{h}_{i-1}$ as input and generates the new hidden embedding $\boldsymbol{h}_i$. This new hidden embedding represents an updated summary of the past events including $x_i$. Formally,

$$\boldsymbol{h}_0 = \boldsymbol{0} \text{ and } \boldsymbol{h}_i = \psi(x_i, \boldsymbol{h}_{i-1}), \quad i = 1, 2, \ldots, N_T.$$

We denote the parameters of both models $g$ and $\psi$ using $\theta \in \Theta$. The model architecture is presented in Figure 1 (a).

**Connection to marked temporal point processes** The proposed framework draws its statistical inspiration from MTPPs. Unlike other recent attempts at modeling point processes, our framework *approximates the conditional probability of events using generated samples* rather than directly specifying the conditional intensity in (1) or PDF in (2) using a parametric model (Du et al., 2016; Mei & Eisner, 2017; Omi et al., 2019; Shchur et al., 2020; Zhu et al., 2022; 2021b).

As illustrated by Figure 1 (b), when our model generates an event denoted by $\widetilde{x} = (t + \Delta\widetilde{t}, \widetilde{m})$, it implies that the resulting event $\widetilde{x}$ follows a conditional probabilistic distribution that is determined by the model parameter $\theta$ and the event's history $\mathcal{H}_t$:

$$\widetilde{x} \sim f_\theta(x|\mathcal{H}_{t(x)}),$$

where $f_\theta$ denotes the conditional PDF of the underlying MTPP (2). This design has three main advantages compared to other point process models:

1. *Generative efficiency*: The generative nature of our model confers an exceptional efficiency in simulating a complete event series for any point processes without relying on thinning algorithms (Ogata, 1981). To exemplify, thinning algorithm (Algorithm 4) has a time complexity of $\mathcal{O}(N^d \cdot N_T)$ to generate $N_T$ events from a history-dependent point process in $d$-dimensional space $\mathcal{X}$, with $N \gg N_T$ being the number of uniformly sampled candidates in one dimension. In contrast, our generation process (Algorithm 1) only requires a complexity of $\mathcal{O}(N_T)$.

2. *Expressiveness*: The proposed model enjoys considerable representational power, as it does not impose any restrictions on the parametric form of the conditional intensity $\lambda$ or PDF $f$. The numerical findings also indicate that our model is capable of capturing complex event interactions, even in a multi-dimensional space.

3. *Predictive efficiency*: To predict the next event $\widehat{x}_i = (t_{i-1} + \Delta\widehat{t}_i, \widehat{m}_i)$ given the observed events' history $\mathcal{H}_{t_i}$, we can calculate the sample average over a set of generated events $\{\widetilde{x}_i^{(l)}\}$ without the need for an explicit expectation computation, *i.e.*,

$$\widehat{x}_i = \int_{(t_{i-1}, +\infty) \times \mathcal{M}} x \cdot f_\theta(x|\mathcal{H}_{t(x)})dx \approx \frac{1}{L}\sum_{l=1}^{L} \widetilde{x}_i^{(l)},$$

---

**Algorithm 1** Event generation process using `CEG`

---

**Input:** Generator $g$, history encoder $\psi$, time horizon $T$
**Initialization:** $\mathcal{H}_T = \emptyset, \boldsymbol{h}_0 = \boldsymbol{0}, t = 0, i = 0$
**while** $t < T$ **do**
    1. Sample $z \sim \mathcal{N}(0, I)$;
    2. Generate next event $\widetilde{x} = (t + \Delta\widetilde{t}, \widetilde{m})$, where $(\Delta\widetilde{t}, \widetilde{m}) = g(z, \boldsymbol{h}_i)$;
    3. $i = i + 1; t = t + \Delta\widetilde{t}; x_i = \widetilde{x}; \mathcal{H}_T = \mathcal{H}_T \cup \{x_i\}$;
    4. Update hidden embedding $\boldsymbol{h}_i = \psi(x_i, \boldsymbol{h}_{i-1})$;
**end while**
**if** $t(x_i) \geq T$ **then**
    **return** $\mathcal{H}_T - \{x_i\}$
**else**
    **return** $\mathcal{H}_T$
**end if**

---

where $L$ denotes the number of samples.

## 2.3 MODEL ESTIMATION

To learn the model, one can maximize the log-likelihood of the observed event series (3). An equivalent form of this objective can be expressed using conditional PDF, as shown in the following equation (see Appendix A for the derivation):

$$\max_{\theta \in \Theta} \; \ell(\theta) := \frac{1}{K} \sum_{k=1}^{K} \int_{\mathcal{X}} \log f_\theta(x | \mathcal{H}_{t(x)}) \, d\mathbb{N}_k(x), \tag{5}$$

where $K$ represents the total number of observed event sequences and $\mathbb{N}_k$ is the counting measure of the $k$-th event sequence. It is worth noting that this learning objective circumvents the need to compute the integral in the second term of (3), which can be computationally intractable when events exist in a multi-dimensional data space.

Now the key challenge is *how do we obtain the conditional PDF of an event $x$ without access to the function $f_\theta$?* This is a commonly posed inquiry in the realm of generative model learning, and there are several pre-existing learning algorithms intended for generative models that can provide solutions to this question (Bond-Taylor et al., 2021). In the rest of this section, we present two learning strategies that approximate the conditional PDF using generated samples and demonstrate the effectiveness of the proposed approach using numerical examples.

**Non-parametric density estimation** We present a non-parametric learning strategy that approximates the conditional PDF using kernel density estimation (KDE). Specifically, the conditional PDF of the $i$-th event $x_i$ can be estimated by,

$$f_\theta(x_i | \mathcal{H}_{t_i}) \approx \frac{1}{L} \sum_{l=1}^{L} \kappa_\sigma(x_i - \widetilde{x}_i^{(l)}), \tag{6}$$

where $\{\widetilde{x}_i^{(l)}\}_{l=1}^{L}$ is a set of samples generated by model $g(\cdot, \boldsymbol{h}_{i-1})$ and $\kappa_\sigma$ is a kernel function with a bandwidth $\sigma$. See our implementation details in Appendix B.

We note that it is important to consider boundary correction (Jones, 1993) for the kernel function in the time dimension, as the support of the next event's time is $[0, +\infty)$, and a regular KDE would extend it to negative infinity. To select the kernel bandwidth $\sigma$, we adopt a common approach called the *self-tuned kernel* (Cheng & Wu, 2022; Mall et al., 2013). This method dynamically determines a value of $\sigma$ for each sample $\widetilde{x}^{(j)}$ by computing the $k$-nearest neighbor ($k$NN) distance among other generated samples. The use of self-tuned kernels is crucial for the success of the model because the event distribution may change significantly over the training iterations. Therefore, adapting the bandwidth for each iteration and sample is necessary to achieve an accurate estimate of the conditional PDF.

**Variational approximation** Variational method is another widely-adopted approach for learning a wide spectrum of generative models. Examples of such models include variational autoencoders

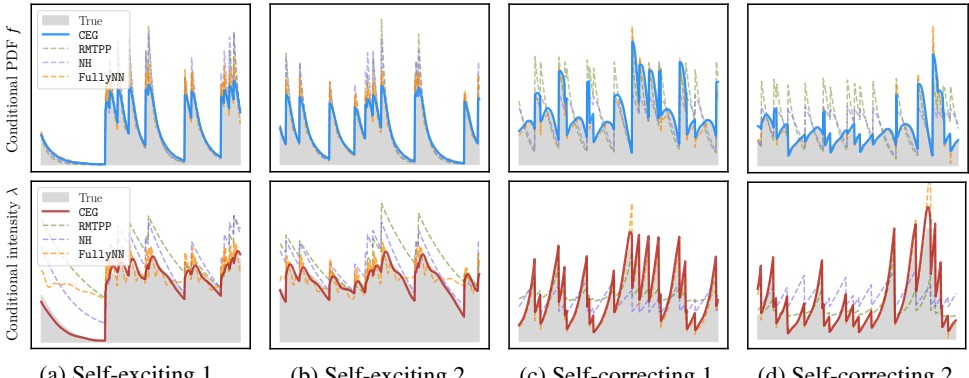

(a) Self-exciting 1   (b) Self-exciting 2   (c) Self-correcting 1   (d) Self-correcting 2

Figure 2: Out-of-sample estimation of the conditional PDF $f(t|\mathcal{H}_t)$ and the corresponding intensity $\lambda(t|\mathcal{H}_t)$ using the proposed method on one-dimensional (time only) synthetic event sequences.

(Kingma & Welling, 2014; Kingma et al., 2019) and diffusion models (Ho et al., 2020; Kingma et al., 2021; Sohl-Dickstein et al., 2015a). In this paper, we follow the idea of conditional variational autoencoder (CVAE) (Sohn et al., 2015) and approximate the log conditional PDF using its evidence lower bound (ELBO):

$$\log f_\theta(x_i|\mathcal{H}_{t_i}) \geq -D_{\mathrm{KL}}(q(z|x_i, \boldsymbol{h}_{i-1})||p_\theta(z|\boldsymbol{h}_{i-1})) + \mathbb{E}_{q(z|x_i, \boldsymbol{h}_{i-1})}\left[\log p_\theta(x_i|z, \boldsymbol{h}_{i-1})\right], \quad (7)$$

where $q$ is a variational approximation of the posterior distribution over the random noise given observed $i$-th event $x_i$ and its history $\boldsymbol{h}_{i-1}$. The first term on the right-hand side is the Kullback–Leibler (KL) divergence of the approximate posterior $q(\cdot|x_i, \boldsymbol{h}_{i-1})$ from the exact posterior $p_\theta(\cdot|\boldsymbol{h}_{i-1})$. The second term is the log-likelihood of the latent data generating process. The complete derivation of (7) and implementation details can be found in the Appendix C.

## 3 EXPERIMENTS

We evaluate our method using both synthetic and real data and demonstrate the superior performance compared to five state-of-the-art approaches, including (1) Recurrent marked temporal point processes (RMTPP) Du et al. (2016), (2) Neural Hawkes (NH) (Mei & Eisner, 2017), (3) Fully neural network based model (FullyNN) (Omi et al., 2019), (4) Epidemic type aftershock sequence (ETAS) (Ogata, 1998) model, (5) Deep non-stationary kernel in point process (DNSK) (Dong et al., 2022). The first three baselines leverage neural networks to model temporal event data (or only with categorical marks). The last two baselines are chosen for testing multi-dimensional event data. Meanwhile, the DNSK is the state-of-the-art method that uses neural networks for high-dimensional mark modeling. See Shchur et al. (2021) and Appendix E for a detailed review of these baseline models. In the following, we refer to our proposed method as the conditional event generator (CEG). Details about the experimental setup and our model architecture are presented in Appendix E.

### 3.1 SYNTHETIC DATA

We first evaluate our model on synthetic data. To be specific, we generate four one-dimensional (1D) and a three-dimensional (3D) synthetic data sets: Four 1D (time only) data sets include 1,000 sequences each, with an average length of 135 events per sequence, and are simulated by two self-exciting processes and two self-correcting processes, respectively, using thinning algorithm (Algorithm 4 in Appendix E). The 3D (time and space) data set also includes 1,000 sequences, each with an average length of 150, generated by a randomly initialized CEG using Algorithm 1.

Table 1: Performance comparison with five baseline methods.

| Model | 1D self-exciting data | | | 1D self-correcting data | | | 3D synthetic data | | | 3D earthquake data |
|---|---|---|---|---|---|---|---|---|---|---|
| | Testing $\ell$ | MRE of $f$ | MRE of $\lambda$ | Testing $\ell$ | MRE of $f$ | MRE of $\lambda$ | Testing $\ell$ | MRE of $f$ | MRE of $\lambda$ | Testing $\ell$ |
| RMTPP | −1.051 (0.015) | 0.437 | 0.447 | −0.975 (0.006) | 0.308 | 0.391 | / | / | / | / |
| NH | −0.776 (0.035) | 0.175 | 0.198 | −1.004 (0.010) | 0.260 | 0.363 | / | / | / | / |
| FullyNN | −1.025 (0.003) | 0.233 | 0.330 | −0.821 (0.008) | 0.322 | 0.495 | / | / | / | / |
| ETAS | / | / | / | / | / | / | −4.832 (0.002) | 0.981 | 0.902 | −3.939 (0.002) |
| DNSK | −0.649 (0.002) | 0.015 | **0.024** | −2.832 (0.004) | 0.134 | 0.185 | −2.560 (0.004) | 0.339 | 0.415 | −3.606 (0.003) |
| **CEG** | **−0.645** (0.002) | **0.013** | 0.066 | **−0.768** (0.005) | **0.042** | **0.075** | **−2.540** (0.011) | **0.049** | **0.089** | **−2.629** (0.015) |

*Numbers in parentheses present standard error for three independent runs.

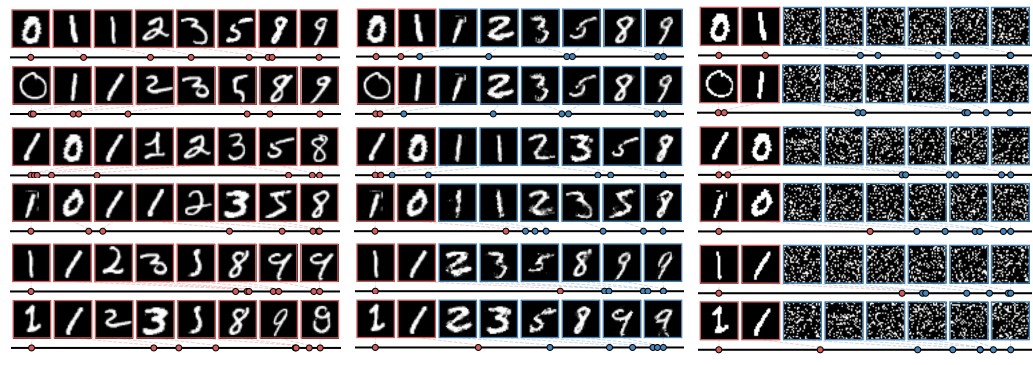

(a) True series      (b) `CEG` generated series      (c) `DNSK` generated series

Figure 3: Generated T-MNIST series using `CEG` and a neural point process baseline `DNSK`, with true sequences displayed on the left. Each event series is generated (blue boxes) given the first two true events (red boxes).

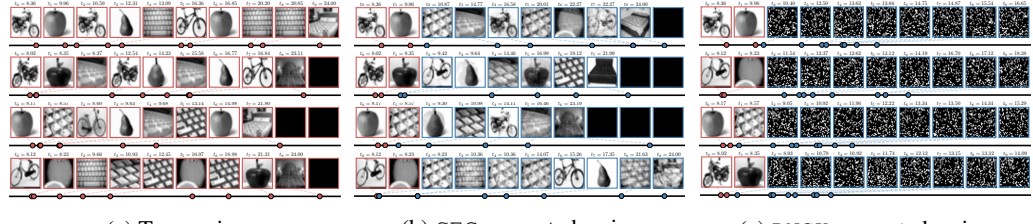

(a) True series      (b) `CEG` generated series      (c) `DNSK` generated series

Figure 4: Generated T-CIFAR series using `CEG` and a neural point process baseline `DNSK`, with true sequences displayed on the left. Each event series is generated (blue boxes) given the first two true events (red boxes).

To assess the effectiveness of our model, we computed the mean relative error (MRE) of the estimated conditional intensity and PDF on the testing set, and compared them to the ground truth. To obtain the conditional intensity and log-likelihood for our method, we can follow Appendix A and (5), respectively, based on the estimated conditional PDF. However, for the baseline approaches, we can only directly estimate the conditional intensity and must compute the corresponding conditional PDF and log-likelihood by numerical approximations using (2) and (3), respectively.

Figure 2 shows the estimated conditional PDFs and intensities, as well as their corresponding ground truth on four 1D testing sequences. The two sequences on the left are generated by a self-exciting process, while the other two on the right are generated by a self-correcting process. Our generative model, `CEG`, enjoys demonstrated superior performance compared to other baseline models in accurately recovering the true conditional PDFs and intensities for both sets of data. Table 1 presents more quantitative results on 1D and 3D data sets, including log-likelihood testing per events and the mean relative error (MRE) of the recovered conditional density and intensity. These results demonstrate the consistent superiority of `CEG` over other methods across all scenarios. Figure E3 in Appendix E presents similar results of the estimated conditional probability density on a 3D synthetic data set, where `CEG` accurately captures the complex spatio-temporal point patterns while `DNSK` and `ETAS` fail to do so.

## 3.2 SEMI-SYNTHETIC DATA WITH IMAGE MARKS

We test our model's capability of generating complex high-dimensional marked events on two semi-synthetic data, including time-stamped MNIST (T-MNIST) and CIFAR-100 (T-CIFAR). In these data sets, both the mark (the image category) and the timestamp are generated through a marked point process. Images from MNIST and CIFAR-100 are subsequently chosen at random based on these marks, acting as an high-dimensional representation of the original image category. It's important to note that during the training phase, categorical marks are excluded, retaining only the high-dimensional images for model learning. Since calculating the log-likelihood for event series with high-dimensional marks is infeasible for `CEG` (the number of samples needed to estimate density is impractically large), we evaluate the model performance according to: (1) the quality of the generated

image marks and (2) the transition dynamics of the entire series. Details of the data generation processes can be found in Appendix E.

1. T-MNIST: For each sequence in the data, the actual digit in the succeeding image is the aggregate of the digits in the two preceding marks. The initial two digits are randomly selected from 0 and 1. The digits in the marks must be less than nine. The hand-written image for each mark is then chosen from the corresponding subset of MNIST according to the digit. The time for the entire MNIST series conforms to a Hawkes process with an exponentially decaying kernel.

2. T-CIFAR: The data contains event series that depict a typical day in the life of a graduate student, spanning from 8:00 to 24:00. The marks are sampled from four categories: outdoor exercises, food ingestion, working, and sleeping. Depending on the most recent activity, the subsequent one is determined by a transition probability matrix. Images are selected from the respective categories to symbolize each activity. The time for these activities follows a self-correcting process.

Figure 3 presents the true T-MNIST series alongside the series generated by `CEG` and `DNSK` given the first two events. Our model not only generates high-dimensional event marks that resemble true images, but also successfully captures the underlying data dynamics, such as the clustering patterns of the self-exciting process and the transition pattern of image marks. On the contrary, the `DNSK` only learns the temporal effects of historical events and struggles to estimate the conditional intensity for the high-dimensional marks. Besides, the grainy images generated by `DNSK` demonstrate the challenge of simulating credible high-dimensional content using thinning algorithm. This is because the real data points, being sparsely scattered in the high-dimensional mark space, make it challenging for the candidate points to align closely with them in the thinning algorithm.

Similar results are shown in Figure 4 on T-CIFAR data set, where the `CEG` is able to simulate high-quality daily activities with high-dimensional content at appropriate times. However, the `DNSK` fails to extract any meaningful patterns from the data, since intensity-based modeling and data generation become ineffectual in high-dimensional mark space.

### 3.3 REAL DATA

In our real data results, our model demonstrates superior efficacy in generating multi-dimensional event sequences of high quality, which closely resemble real event series.

**Northern California earthquake catalog**    We test our method using the Northern California Earthquake Data (Northern California Earthquake Data Center. UC Berkeley Seismological Laboratory. Dataset, 2014), which contains detailed information on the timing and location of earthquakes that occurred in central and northern California from 1978 to 2018, totaling 5,984 records with magnitude greater than 3.5. We divided the data into several sequences by month. In comparison to other baseline methods that can only handle 1D event data, we primarily evaluated our model against `DNSK` and `ETAS`, training each model using 80% of the dataset and testing them on the rest. To demonstrate the effectiveness of our method on the real data, we assess the quality of the generated sequences by each model. Our model's generation process for new sequences can be efficiently carried out using Algorithm 1, whereas both `DNSK` and `ETAS` requires the use of a thinning algorithm (Algorithm 4) for simulation. We also compared the estimated conditional probability density functions (PDFs) of real sequences by each model in Appendix E.

We compare the generative ability of each method in Figure 5. The top left sub-figure features a single event series selected at random from the data set, while the rest of the sub-figures in the first row exhibit event series generated by each model, respectively. The quality of the generated earthquake sequence using our method is markedly superior to that generated by `DNSK` and `ETAS`. We also simulate multiple sequences using each method and visualize the spatial distribution of generated earthquakes in the second row. The shaded area reflects the spatial density of earthquakes obtained by KDE and represents the "background rate" over space. It is evident that `CEG` is successful in capturing the underlying earthquake distribution, while the two STPP baselines are unable to do so. Additional results in Figure E5 visualizes the conditional PDF estimated by `CEG`, `DNSK`, and `ETAS` for an actual earthquake sequence in testing set, respectively. The results indicate that our model is able to capture the heterogeneous triggering effects among earthquakes which align with current understandings of the San Andreas Fault System (Wallace, 1990). However, both `DNSK` and `ETAS` fail to extract this geographical feature from the data.

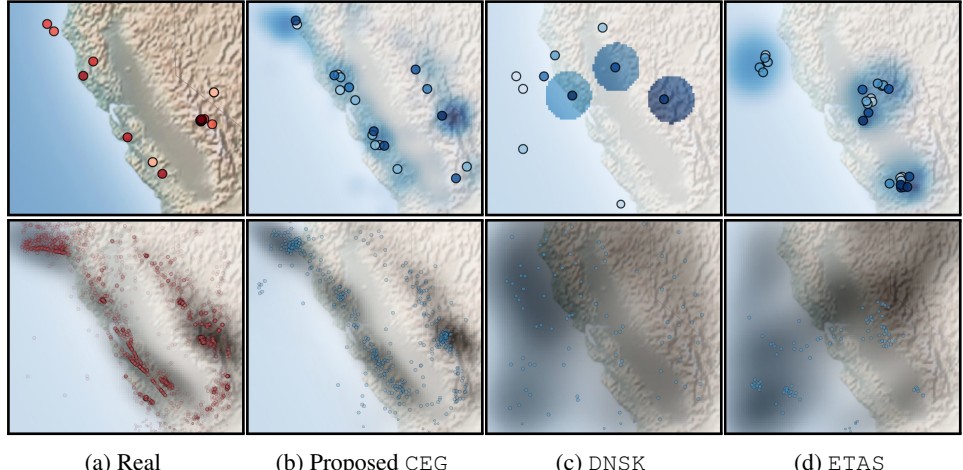

| (a) Real | (b) Proposed `CEG` | (c) `DNSK` | (d) `ETAS` |

Figure 5: Comparison between real and generated earthquake sequence. The first row displays a single sequence, either real or generated, with the color depth of the dots reflecting the occurrence time of each event. Darker colors represent more recent events. The shaded areas represent the estimated conditional PDFs. The second row shows 1,000 real or generated events, where the gray area indicates the high density of events, which can be interpreted as the "background rate".

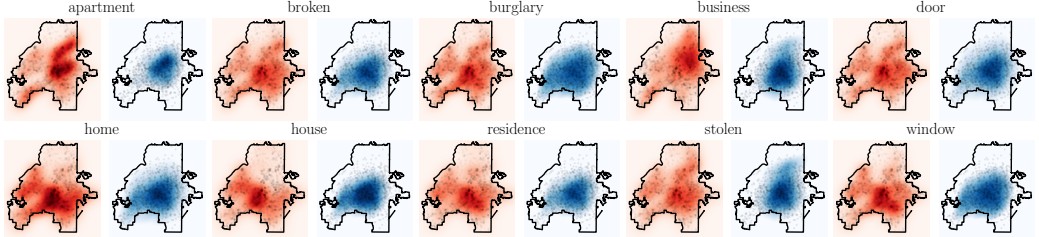

Figure 6: The spatial distributions of the TF-IDF values of 10 crime-related keywords. The heatmap in red and blue represent distributions of TF-IDF value of the keywords in the true and generated events, respectively. The black dots pinpoint the locations of the corresponding events.

**Atlanta crime reports with textual description** We further assess our method using 911-calls-for-service data in Atlanta. The proprietary data set contains 4644 burglary incidents from 2016 to 2017, detailing the time, location, and a comprehensive textual description of each incident. Each textual description was transformed into a TF-IDF vector (Aizawa, 2003), from which the top 10 keywords with the most significant TF-IDF values were selected. The location combined with the corresponding 10-dimensional TF-IDF vector is regarded as the mark of the incident. We first fit our `CEG` model using the preprocessed data, subsequently generate crime event sequences, and then compare them with the real data.

Figure 6 visualizes the spatial distributions of the true and the generated TF-IDF value of each keyword, respectively, signifying the heterogeneous crime patterns across the city. As we can observe, our model is capable of capturing such spatial heterogeneity for different keywords and simulating crime incidents that follow the underlying spatio-temporal-textual dynamics existing in criminological *modus operandi* (Zhu & Xie, 2022).

## 4 DISCUSSIONS

In this paper, we introduce a novel framework for high-dimensional marked temporal point processes for generating high-quality event series, which offers a highly adaptable and efficient solution for modeling and generating multi-dimensional event data. The proposed framework uses a conditional generator to explore the intricate multi-dimensional event space and generates subsequent events based on prior observations with exceptional efficiency. The empirical evaluation demonstrates the superior performance of our model in capturing complex data distribution and generating high-quality samples against state-of-the-art methods, and its flexibility of being adapting to different real-world scenarios.

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
