## A   DERIVATION OF THE CONDITIONAL PROBABILITY OF POINT PROCESSES

The conditional probability of point processes can be derived from the conditional intensity (1). Suppose we are interested in the conditional probability of events at a given point $x \in \mathcal{X}$, and we assume that there are $i$ events that happen before $t(x)$. Let $\Omega(x)$ be a small neighborhood containing $x$. According to (1), we can rewrite $\lambda(x|\mathcal{H}_{t(x)})$ as following:

$$
\begin{aligned}
\lambda(x|\mathcal{H}_{t(x)}) &= \mathbb{E}\left(d\mathbb{N}(x)|\mathcal{H}_{t(x)}\right)/dx = \mathbb{P}\{x_{i+1} \in \Omega(x)|\mathcal{H}_{t(x)}\}/dx \\
&= \mathbb{P}\{x_{i+1} \in \Omega(x)|\mathcal{H}_{t_{i+1}} \cup \{t_{i+1} \geq t(x)\}\}/dx \\
&= \frac{\mathbb{P}\{x_{i+1} \in \Omega(x), t_{i+1} \geq t(x)|\mathcal{H}_{t_{i+1}}\}/dx}{\mathbb{P}\{t_{i+1} \geq t(x)|\mathcal{H}_{t_{i+1}}\}}.
\end{aligned}
$$

Here $\mathcal{H}_{t_{i+1}} = \{x_1, \ldots, x_i\}$ represents the history up to $i$-th events. If we let $F(t(x)|\mathcal{H}_{t(x)}) = \mathbb{P}(t_{i+1} < t(x)|\mathcal{H}_{t_{i+1}})$ be the conditional cumulative probability, and $f(x|\mathcal{H}_{t(x)}) \triangleq f(x_{i+1} \in \Omega(x)|\mathcal{H}_{t_{i+1}})$ be the conditional probability density of the next event happening in $\Omega(x)$. Then the conditional intensity can be equivalently expressed as

$$
\lambda(x|\mathcal{H}_{t(x)}) = \frac{f(x|\mathcal{H}_{t(x)})}{1 - F(t(x)|\mathcal{H}_{t(x)})}.
$$

We multiply the differential $dx = dtdm$ on both sides of the equation and integral over the mark space $\mathcal{M}$:

$$
\begin{aligned}
dt \cdot \int_{\mathcal{M}} \lambda(x|\mathcal{H}_{t(x)})dm &= \frac{dt \cdot \int_{\mathcal{M}} f(x|\mathcal{H}_{t(x)})dm}{1 - F(t(x)|\mathcal{H}_{t(x)})} = \frac{dF(t(x)|\mathcal{H}_{t(x)})}{1 - F(t(x)|\mathcal{H}_{t(x)})} \\
&= -d\log\left(1 - F(t(x)|\mathcal{H}_{t(x)})\right).
\end{aligned}
$$

Hence, integrating over $t$ on $[t_i, t(x))$ leads to the fact that

$$
\begin{aligned}
F(t(x)|\mathcal{H}_{t(x)}) &= 1 - \exp\left(-\int_{t_i}^{t(x)} \int_{\mathcal{M}} \lambda(x|\mathcal{H}_{t(x)})dmdt\right) \\
&= 1 - \exp\left(-\int_{[t_i,t(x))\times\mathcal{M}} \lambda(x|\mathcal{H}_{t(x)})dx\right)
\end{aligned}
$$

because $F(t_i) = 0$. Then we have

$$
f(x|\mathcal{H}_{t(x)}) = \lambda(x|\mathcal{H}_{t(x)}) \cdot \exp\left(-\int_{[t_i,t(x))\times\mathcal{M}} \lambda(x|\mathcal{H}_{t(x)})dx\right),
$$

which corresponds to (2).

The log-likelihood of one observed event series in (3) is derived, by the chain rule, as

$$
\begin{aligned}
\ell(x_1, \ldots, x_{N_T}) = \log f(x_1, \ldots, x_{N_T}) &= \log\prod_{i=1}^{N_T} f(x_i|\mathcal{H}_{t_i}) \\
&= \int_{\mathcal{X}} \log f(x|\mathcal{H}_{t(x)})d\mathbb{N}(x) \\
&= \int_{\mathcal{X}} \log \lambda(x|\mathcal{H}_{t(x)})d\mathbb{N}(x) - \int_{\mathcal{X}} \lambda(x|\mathcal{H}_{t(x)})dx.
\end{aligned}
$$

The log-likelihood of $K$ observed event sequences in (5) can be conveniently obtained with the counting measure $\mathbb{N}$ replaced by the counting measure $\mathbb{N}_k$ for the $k$-th sequence.

## B   IMPLEMENTATION DETAILS OF NON-PARAMETRIC LEARNING

Estimating the conditional PDF $f(x|\mathcal{H}_{t(x)})$ using kernel density estimation (KDE) within our framework presents two main challenges: (1) The distribution density of events generated by certain

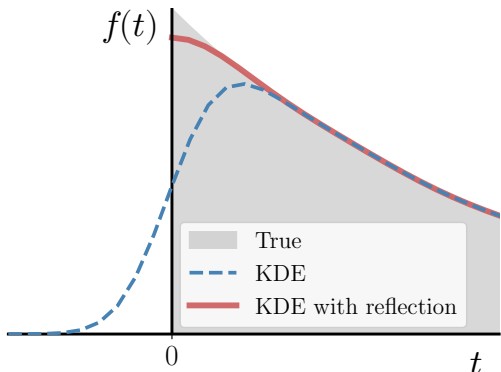

Figure A1: A comparison between the vanilla KDE and the KDE with boundary correction. The grey shaded area indicates the true density function, which is defined on the bounded region $[0, +\infty)$. The blue dashed line and red line show the estimated density function by the vanilla KDE and the KDE with reflection, respectively.

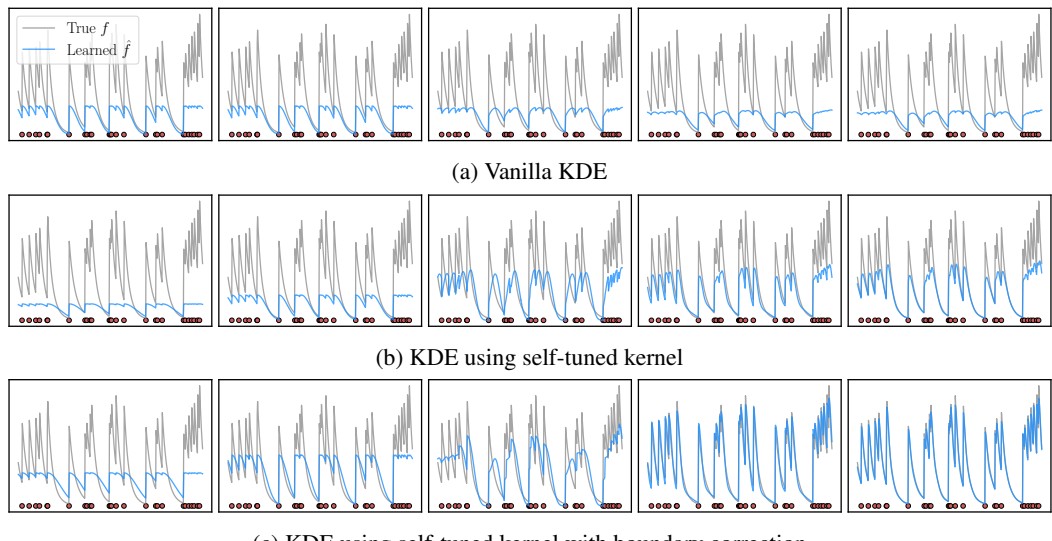

Figure A2: The estimated conditional PDF $f(t|\mathcal{H}_t)$ of a testing sequence is displayed from left to right. Each panel within the same row represents the estimated conditional PDF at intervals of 10 training epochs.

inhomogeneous point processes can vary from location to location in the event space. Consequently, using a single bandwidth for estimation would either oversmooth the conditional PDF or introduce excessive noise in areas with sparse events. (2) The time intervals of the next events are usually clustered in a small neighborhood of 0 and always positive, which will lead to a significant boundary bias.

To overcome the above challenges, we adopt the self-tuned kernel with boundary correction:

1. We first choose the bandwidth adaptively, where the bandwidth $\sigma$ tends to be small for those samples falling into event clusters and to be large for those isolated samples. We dynamically determine the value of $\sigma$ for each sample $\widetilde{x}$ by computing the $k$-nearest neighbor ($k$NN) distance among other generated samples Cheng & Wu (2022); Mall et al. (2013).
2. We correct the boundary bias of KDE by reflecting the data points against the boundary 0 in time domain Jones (1993). Specifically, the kernel with reflection is defined as follows:

$$\kappa(x - \widetilde{x}) = \upsilon^*(\Delta t - \Delta \widetilde{t}) \cdot \upsilon(m - \widetilde{m}),$$

---

**Algorithm 2** Non-parametric learning for `CEG`

---

**Input:** Training set $X$ with $K$ sequences: $X = \{x_i^{(k)}\}_{i=1,\dots,\mathbb{N}_k(\mathcal{X}),\ k=1,\dots,K}$, learning epoch $E$, learning rate $\gamma$, mini-batch size $M$.
**Initialization:** model parameters $\theta$, $e = 0$
**while** $e < E$ **do**
    **for** each sampled batch $\widehat{X}^M$ with size $M$ **do**
        1. Draw samples $z$ from noise distribution $\mathcal{N}(0,1)$;
        2. Feed $z$ into the generator $g$ to obtain sampled events $\widetilde{x}$;
        3. Estimate conditional PDF using KDE (6) and log-likelihood $\ell$ (3), given data $\widehat{X}^M$, samples $\widetilde{x}$ and the model;
        4. $\theta \leftarrow \theta + \gamma \partial \ell / \partial \theta$;
    **end for**
    $e \leftarrow e + 1$;
**end while**
**return** $\theta$

---

where $\upsilon$ is an arbitrary kernel and $\upsilon^*(x - \widetilde{x}) = \upsilon(x - \widetilde{x}) + \upsilon(-x - \widetilde{x})$ is the same kernel with reflection boundary. This allows for a more accurate estimation of the density near the boundary of the time domain without impacting the estimation elsewhere, as shown in Figure A1.

Figure A2 compares the learned conditional PDF using three KDE methods on the same synthetic data set generated by a self-exciting Hawkes process. The results show that the estimation using the self-tuned kernel with boundary correction shown in (c) significantly outperforms two ablation models in (a) and (b). We also summarize the learning algorithm in Algorithm 2.

## C   DERIVATION AND IMPLEMENTATION DETAILS OF VARIATIONAL LEARNING

**Derivation of the approximate conditional PDF**   Now we present the derivation of the approximate conditional PDF in (7). We first use hidden embedding $\boldsymbol{h}$ to represent the history $\mathcal{H}_t(x)$ and $f_\theta(x|\mathcal{H}_{t(x)})$ can be substituted by $f_\theta(x|\boldsymbol{h})$. Then the conditional PDF of event $x$ given the history can be re-written as:

$$\log f_\theta(x|\boldsymbol{h}) = \log \int p_\theta(x, z|\boldsymbol{h}) dz,$$

where $z$ is a latent random variable. This integral has no closed form and can usually be estimated by Monte Carlo integration with importance sampling, *i.e.*,

$$\int p_\theta(x, z|\boldsymbol{h}) dz = \mathbb{E}_{z \sim q(\cdot|x, \boldsymbol{h})} \left[ \frac{p_\theta(x, z|\boldsymbol{h})}{q(z|x, \boldsymbol{h})} \right].$$

Here $q(z|x, \boldsymbol{h})$ is the proposed variational distribution, where we can draw sample $z$ from this distribution given $x$ and $\boldsymbol{h}$. Therefore, by Jensen's inequality, we can find the evidence lower bound (ELBO) of the conditional PDF:

$$\log f_\theta(x|\boldsymbol{h}) = \log \mathbb{E}_{z \sim q(\cdot|x, \boldsymbol{h})} \left[ \frac{p_\theta(x, z|\boldsymbol{h})}{q(z|x, \boldsymbol{h})} \right] \geq \mathbb{E}_{z \sim q(\cdot|x, \boldsymbol{h})} \left[ \log \frac{p_\theta(x, z|\boldsymbol{h})}{q(z|x, \boldsymbol{h})} \right].$$

Using Bayes rule, the ELBO can be equivalently expressed as:

$$
\begin{aligned}
\mathbb{E}_{z \sim q(\cdot|x, \boldsymbol{h})} \left[ \log \frac{p_\theta(x, z|\boldsymbol{h})}{q(z|x, \boldsymbol{h})} \right] &= \mathbb{E}_{z \sim q(\cdot|x, \boldsymbol{h})} \left[ \log \frac{p_\theta(x|z, \boldsymbol{h}) p_\theta(z|\boldsymbol{h})}{q(z|x, \boldsymbol{h})} \right] \\
&= \mathbb{E}_{z \sim q(\cdot|x, \boldsymbol{h})} \left[ \log \frac{p_\theta(z|\boldsymbol{h})}{q(z|x, \boldsymbol{h})} \right] + \mathbb{E}_{z \sim q(\cdot|x, \boldsymbol{h})} \left[ \log p_\theta(x|z, \boldsymbol{h}) \right] \\
&= -D_{\mathrm{KL}}(q(z|x, \boldsymbol{h}) || p_\theta(z|\boldsymbol{h})) + \mathbb{E}_{z \sim q(\cdot|x, \boldsymbol{h})} \left[ \log p_\theta(x|z, \boldsymbol{h}) \right].
\end{aligned}
$$

**Implementation details**   In practice, we introduce two additional generator functions, *encoder net* $g_{\mathrm{encode}}(\epsilon, x_i, \boldsymbol{h}_{i-1})$ and *prior net* $g_{\mathrm{prior}}(\epsilon, \boldsymbol{h}_{i-1})$, respectively, to represent $q(z|x_i, \boldsymbol{h}_{i-1})$ and $p_\theta(z|\boldsymbol{h}_{i-1})$ as transformations of another random variable $\epsilon \sim \mathcal{N}(0, I)$ using reparametrization trick

---

**Algorithm 3** Variational learning for `CEG` using stochastic gradient descent

---

**Input:** Training set $X$ with $K$ sequences: $X = \{x_i^{(k)}\}_{i=1,\ldots,\mathbb{N}_k(\mathcal{X}),\ k=1,\ldots,K}$, learning epoch $E$, learning rate $\gamma$, mini-batch size $M$.
**Initialization:** model parameters $\theta$, $e = 0$
**while** $e < E$ **do**
    **for** each sampled batch $\widehat{X}^M$ with size $M$ **do**
        1. Draw samples $\epsilon$ from noise distribution $\mathcal{N}(0,1)$;
        2. Compute $z$ using reparametrization trick, given data $\widehat{X}^M$, noise $\epsilon$, $g_{\text{prior}}$, and $g_{\text{encode}}$;
        3. Compute ELBO (7) and log-likelihood $\ell$ (3) based on $z$ and data $\widehat{X}^M$;
        4. $\theta \leftarrow \theta + \gamma \partial \ell / \partial \theta$;
    **end for**
    $e \leftarrow e + 1$;
**end while**
**return** $\theta$

---

(Sohl-Dickstein et al., 2015a). Both $q(z|x_i, \boldsymbol{h}_{i-1})$ and $p_\theta(z|\boldsymbol{h}_{i-1})$ are often modeled as Gaussian distributions, which allows us to compute the KL divergence of Gaussians with a closed-form expression. The log-likelihood of the second term can be implemented as the reconstruction loss and calculated using generated samples.

We parameterize both $p_\theta(z|\boldsymbol{h})$ and $q(z|x, \boldsymbol{h})$ using fully-connected neural networks with one hidden layer, denoted by $g_{\text{prior}}$ and $g_{\text{encode}}$, respectively. The prior of the latent variable is modulated by the input $\boldsymbol{h}$ in our formulation; however, the constraint can be easily relaxed to make the latent variables statistically independent of input variables, *i.e.*, $p_\theta(z|\boldsymbol{h}) = p_\theta(z)$ Kingma et al. (2014); Sohn et al. (2015). For the approximate posterior $q(z|x, \boldsymbol{h})$, a common choice is a simple factorized Gaussian encoder, which can be represented as:

$$q(z|x, \boldsymbol{h}) = \mathcal{N}(z; \mu, \text{diag}(\Sigma)),$$

or

$$q(z|x, \boldsymbol{h}) = \prod_{j=1}^{r} q(z_j|x, \boldsymbol{h}) = \prod_{j=1}^{r} \mathcal{N}(z_j; \mu_j, \sigma_j^2).$$

The Gaussian parameters $\mu = (\mu_j)_{j=1,\ldots,r}$ and $\text{diag}(\Sigma) = (\sigma_j^2)_{j=1,\ldots,r}$ are the output of an encoder network $\phi$ and the latent variable $z$ can be obtained using reparametrization trick:

$$(\mu, \log \text{diag}(\Sigma)) = \phi(x, \boldsymbol{h}),$$
$$z = \mu + \text{diag}(\Sigma) \odot \epsilon,$$

where $\epsilon \sim \mathcal{N}(0, I)$ is another random variable and $\odot$ is the element-wise product. For simplicity in presentation, we denote such a factorized Gaussian encoder as $g_{\text{encode}}(\epsilon, x, \boldsymbol{h})$ that maps an event $x$, its history $\boldsymbol{h}$, and a random noise vector $\epsilon$ to a sample $z$ from the approximate posterior for that event $x$.

In (7), the first term is the KL divergence of the approximate posterior from the prior, which acts as a regularizer, while the second term is an expected negative reconstruction error. They can be calculated as follows: (1) Because both $q(z|x_i, \boldsymbol{h}_{i-1})$ and $p_\theta(z|\boldsymbol{h}_{i-1})$ are modeled as Gaussian distributions, the KL divergence can be computed using a closed-form expression. (2) Minimizing the negative log-likelihood $p_\theta(x|z, \boldsymbol{h})$ is equivalent to maximizing the cross entropy between the distribution of an observed event $x$ and the reconstructed event $\widetilde{x}$ generated by the generative model $g$ given $z$ and the history $\boldsymbol{h}$. The learning algorithm has been summarized in Algorithm 3.

## D    SAMPLING EFFICIENCY COMPARISON

Thinning algorithms are known to be challenging and suffer from low sampling efficiency. This is because (i) these algorithms require sampling uniformly in the space $\mathcal{X}$ with the upper limit of the conditional intensity $\overline{\lambda} > \lambda(x)$, $\forall x$, and only a few candidate points are retained in the end. (ii) the decision of whether to reject one candidate point requires the evaluation of the conditional

Table D1: Computation costs for generating earthquake series and time-stamped image series of length 100 using `ETAS`, `DNSK` and `CEG`.

| | 3D earthquake data | | T-MNIST | | T-CIFAR | |
|---|---|---|---|---|---|---|
| Model | 5 sequences | 50 sequences | 5 sequences | 50 sequences | 5 sequences | 50 sequences |
| `ETAS` | 12.4 | 118.6 | / | / | / | / |
| `DNSK` | 20.1 | 220.4 | 87.3 | 745.6 | 274.0 | 1381.9 |
| `CEG` | $< 1$ | $< 1$ | 0.6 | 0.8 | 1.1 | 1.2 |

*Unit: second.

intensity function over the entire history, which is also stochastic. This doubly stochastic trait makes the entire thinning process particularly costly when $\mathcal{X}$ is a multi-dimensional space, since it requires a drastically large number of candidate points and numerous evaluations of the conditional intensity function.

On the contrary, our model generates samples based on the underlying conditional distribution of events learned from true data, thus every generated point will be retained. Table D1 compares the time costs for `ETAS`, `DNSK`, and `CEG` to generate event series of length 100 on each data set. Particularly noteworthy is that our model requires a similar amount of time to generate different numbers of sequences. This is because `CEG` can generate all the sequences in parallel, leveraging the benefits of the implementation of conditional generative models.

## E    EXPERIMENT DETAILS AND ADDITIONAL RESULTS

**Baselines**    We compare our proposed method empirically with the following baselines:

1. *Recurrent Marked Temporal Point Process* (`RMTPP`) Du et al. (2016) uses an RNN to capture the nonlinear relationship between both the markers and the timings of past events. It models the conditional intensity function by

$$\lambda(t|\mathcal{H}_t) = \exp(\boldsymbol{v}^\top \boldsymbol{h}_i + w(t - t_i) + b),$$

where hidden state $\boldsymbol{h}_i$ of the RNN represents the event history until the nearest $i$-th event $\mathcal{H}_{t_i} \cup \{t_i\}$. The $\boldsymbol{v}, w, b$ are trainable parameters. The model is learned by MLE using backpropagation through time (BPTT).

2. *Neural Hawkes Process* (`NH`) Mei & Eisner (2017) extends the classical Hawkes process by memorizing the long-term effects of historical events. The conditional intensity function is given by

$$\lambda(t|\mathcal{H}_t) = f(\boldsymbol{w}^\top \boldsymbol{h}_t),$$

where $\boldsymbol{h}_t$ is a sufficient statistic of the event history modeled by the hidden state in a continuous-time LSTM, and $f(\cdot)$ is a scaled softplus function for ensuring positive output. The weight $\mathbf{w}$ is learned jointly with the LSTM through MLE.

3. *Fully Neural Network based Model* (`FullyNN`) for General Temporal Point Processes Omi et al. (2019) models the cumulative hazard function given the history embedding $\boldsymbol{h}_i$, which leads to a tractable likelihood. It uses a fully-connect neural network $Z_i$ with a non-negative activation function for the cumulative hazard function $\Phi(\tau|\boldsymbol{h}_i)$ where $\tau = t - t_i$. The conditional intensity function is obtained by computing the derivative of the network:

$$\lambda(t|\mathcal{H}_t) = \frac{\partial}{\partial(\tau)} \Phi(\tau|\boldsymbol{h}_i) = \frac{\partial}{\partial(\tau)} Z_i(\tau),$$

where $Z_i$ is the fully-connect neural network.

4. *Epidemic-type aftershock sequence* (`ETAS`) acts as a benchmark in spatio-temporal point process modeling. Denoting each event $x := (t, s)$, `ETAS` adopts a Gaussian diffusion kernel in the conditional intensity as following

$$\lambda(t, s|\mathcal{H}_t) = \mu + \sum_{(t_i, s_i) \in \mathcal{H}_t} k(t, t_i, s, s_i),$$

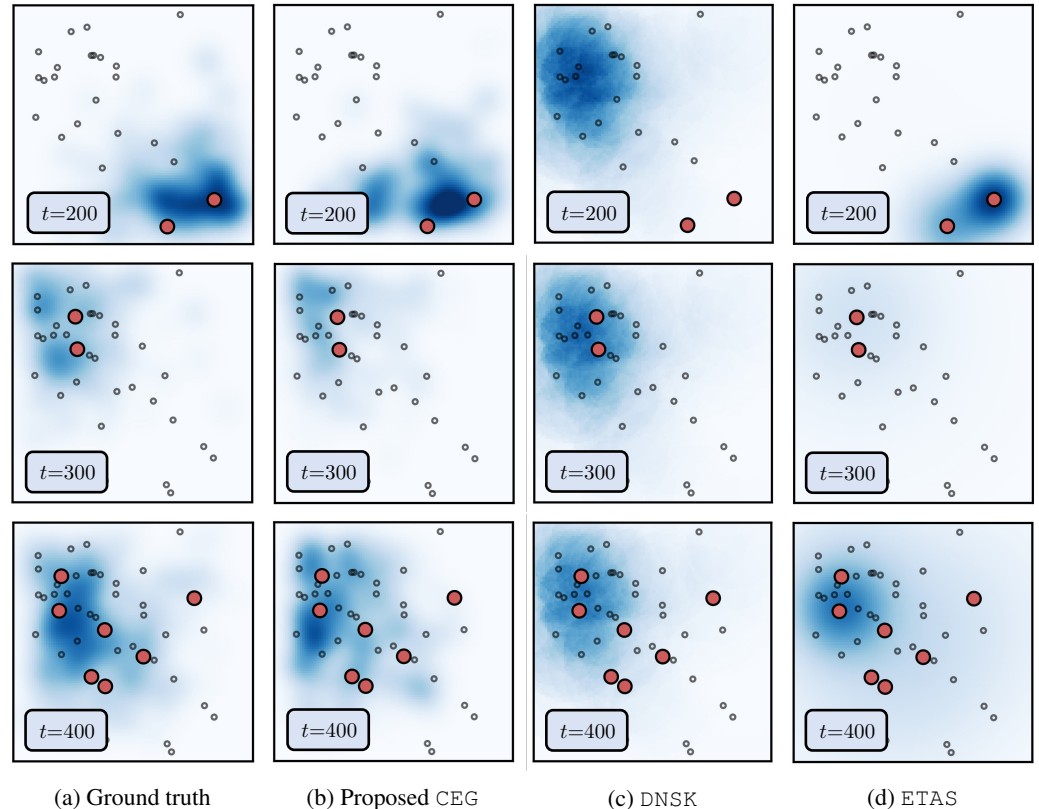

(a) Ground truth     (b) Proposed `CEG`     (c) `DNSK`     (d) `ETAS`

Figure E3: Snapshots of out-of-sample estimation of the conditional PDFs for a three-dimensional (time and space) synthetic event sequence, arranged in chronological order from left to right. The conditional PDFs are indicated by shaded areas, with darker shades indicating higher conditional PDF values. The red dots represent newly observed events within the most recent time period, while the circles represent historical events.

where

$$k(t, t_i, s, s_i) = \frac{Ce^{-\beta(t-t_i)}}{2\pi\sqrt{|\Sigma|}(t-t_i)} \cdot \exp\left\{-\frac{(s-s_i-a)^\top \Sigma^{-1}(s-s_i-a)}{2(t-t_i)}\right\}.$$

Here $\Sigma = \text{diag}(\sigma_x^2, \sigma_y^2)$ is a diagonal matrix representing the covariance of the spatial correlation. Note that the diffusion kernel is stationary and only depends on the spatio-temporal distance between two events. All the parameters are learnable.

5. *Deep non-stationary kernel* (DNSK) proposes a neural-network-based influence kernel based on kernel singular value decomposition for modeling spatio-temporal point process data. In addition, their kernel can be extended to handle high-dimensional marks:

$$k(t_i, t-t_i, s_i, s-s_i, m_i, m) = \sum_{q=1}^{Q}\sum_{r=1}^{R}\sum_{l=1}^{L} \alpha_{lrq}\psi_l(t_i)\varphi_l(t-t_i)u_r(s_i)v_r(s-s_i)g_q(m_i)h_q(m).$$

Here all the basis functions are represented by fully-connected neural networks.

**Synthetic data description** We use the following point process models to generate the one-dimensional synthetic data sets using Algorithm 4:

1. Self-exciting Hawkes process: $\lambda(t) = \mu + \sum_{t_i \in \mathcal{H}_t} \beta e^{-\beta(t-t_i)}$, with $\mu = 0.1, \beta = 0.1$ and $\mu = 0.5, \beta = 1.0$ in self-exciting data 1 and 2, respectively.
2. Self-correcting process: $\lambda(t) = \exp\left(\mu t - \sum_{t_i \in \mathcal{H}_t} \alpha\right)$, with $\mu = 1.0, \alpha = 1.0$ and $\mu = 0.5, \alpha = 0.8$ in self-correcting data 1 and 2, respectively.

---

**Algorithm 4** Thinning algorithm

---

**Input:** Model $\lambda(\cdot)$, time horizon $T$, mark space $\mathcal{M}$, Intensity upper bound $\bar{\lambda}$.
**Initialization:** $\mathcal{H}_T = \emptyset, t = 0, i = 0$
**while** $t < T$ **do**
   1. Sample $u \sim \text{Unif}(0, 1)$.
   2. $t \leftarrow t - \ln u / \bar{\lambda}$.
   3. Sample $m \sim \text{Unif}(\mathcal{M}), D \sim \text{Unif}(0, 1)$.
   4. $\lambda = \lambda(t, m | \mathcal{H}_T)$.
  **if** $D\bar{\lambda} \leq \lambda$ **then**
     $i \leftarrow i + 1; t_i = t, m_i = m$.
     $\mathcal{H}_T \leftarrow \mathcal{H}_T \cup \{(t_i, m_i)\}$.
  **end if**
**end while**
**if** $t_i \geq T$ **then**
  **return** $\mathcal{H}_T - \{(t_i, m_i)\}$
**else**
  **return** $\mathcal{H}_T$
**end if**

---

3. T-MNIST: In the MNIST series, all the digits that are greater than nine will be truncated to nine. The exponentially decaying kernel for the observation times are $k(t, t_i) = \beta e^{-\beta(t-t_i)}, \beta = 0.2$.

4. T-CIFAR: The images of bicycles and motorcycles represent outdoor exercises; the apples, pears, and oranges represent food ingestion; the computer keyboards represent study/working; and the beds represent sleeping. Before 21:00, the activity series progresses with the transition probability matrix between (exercise, food ingestion, working) being

$$P = \begin{pmatrix} 0.0 & 1.0 & 0.0 \\ 0.2 & 0.0 & 0.8 \\ 0.2 & 0.3 & 0.5 \end{pmatrix}.$$

Starting from 21:00, the probability of sleeping increases linearly from 0 to 1 at 23:00. Each series ends with the activity of sleeping. The self-correcting process for event times is set with $\mu = 0.1, \alpha = 0.5$, indicating that each activity will last for a while before the student moves to the next activity (or stays in the current one).

**Experimental setup** We choose our generator $g$ to be a fully-connected neural network with two hidden layers of width 32 with softplus activation function. To guarantee that the generated time interval is always positive, we apply an extra Rectified Linear Unit (ReLU) function for the output of the time dimension in the output layer. We use an LSTM for the history encoder $\psi$. We train our model and other baselines using $90\%$ of the data and test them on the remaining $10\%$ data. To fit the model parameters, we maximize log-likelihood according to (5), and adopt Adam optimizer (Kingma & Ba, 2014) with a learning rate of $10^{-3}$ and a batch size of 32 (event sequences). More details about experimental setup can be found in Appendix E.

For `RMTPP`, `NH` and `FullyNN`, we take the default parameters for model architectures in the original papers, with the dimension of hidden embedding to be 64 for all three models, and a fully-connected neural network with two hidden layers of width 64 for the cumulative hazard function in `FullyNN`. There is no hyperparameter in `ETAS`. All the baselines are trained using the Adam optimizer with a learning rate of $10^{-3}$ and a batch size of 32 for 100 epochs. The experiments are implemented on Google Colaboratory (Pro version) with 12GB RAM and a Tesla T4 GPU.

### E.1 ADDITIONAL EXPERIMENT RESULTS

**3D synthetic data** Each row in Figure E3 displays four snapshots of estimated conditional probability density functions (PDFs) for a particular 3D testing sequence. It is apparent that our model's estimated PDFs closely match the ground truth and accurately capture the complex spatial and temporal point patterns. Conversely, `DNSK` and `ETAS` model for estimating spatio-temporal point processes fails to capture the heterogeneous triggering effects between events, indicating limited practical representational power.

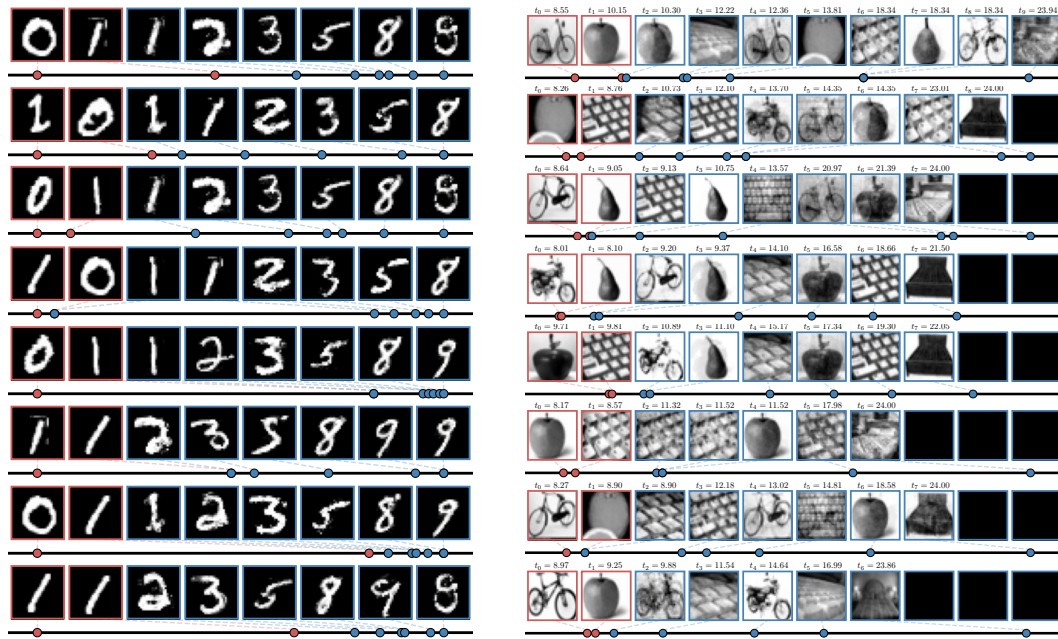

(a) Additional T-MNIST series generated by CEG          (b) Additional T-CIFAR series generated by CEG

Figure E4: Additional T-MNIST and T-CIFAR series using CEG and a neural point process baseline DNSK, with true sequences displayed on the left. Each event series is generated (blue boxes) given the first two true events (red boxes).

**Semi-synthetic image data**   More generated T-MNIST and T-CIFAR series by CEG are presented in Figure E4. As we can see, our generative point process can not only sample images that resemble the ground truth, but also recover the intricate temporal dynamics (*e.g.*, clustering effect of self-exciting process in T-MNIST, student's sleeping time in T-CIFAR) and high-dimensional mark dependencies.

**Northern California earthquake catalog**   Additional results in Figure E5 visualizes the conditional PDF estimated by CEG, DNSK, and ETAS for an actual earthquake sequence in testing set, respectively. The results indicate that our model is able to capture the heterogeneous effects among earthquakes. Particularly noteworthy is our model's finding of a heightened probability of seismic activity along the San Andreas fault, coupled with a diminished likelihood in the basin. These results align with current understandings of the mechanics of earthquakes in Northern California. However, both DNSK and ETAS fail to extract this geographical feature from the data and suggest that observed earthquakes impact their surroundings uniformly, leading to an increased likelihood of aftershocks within a circular area centered on the location of the initial event.

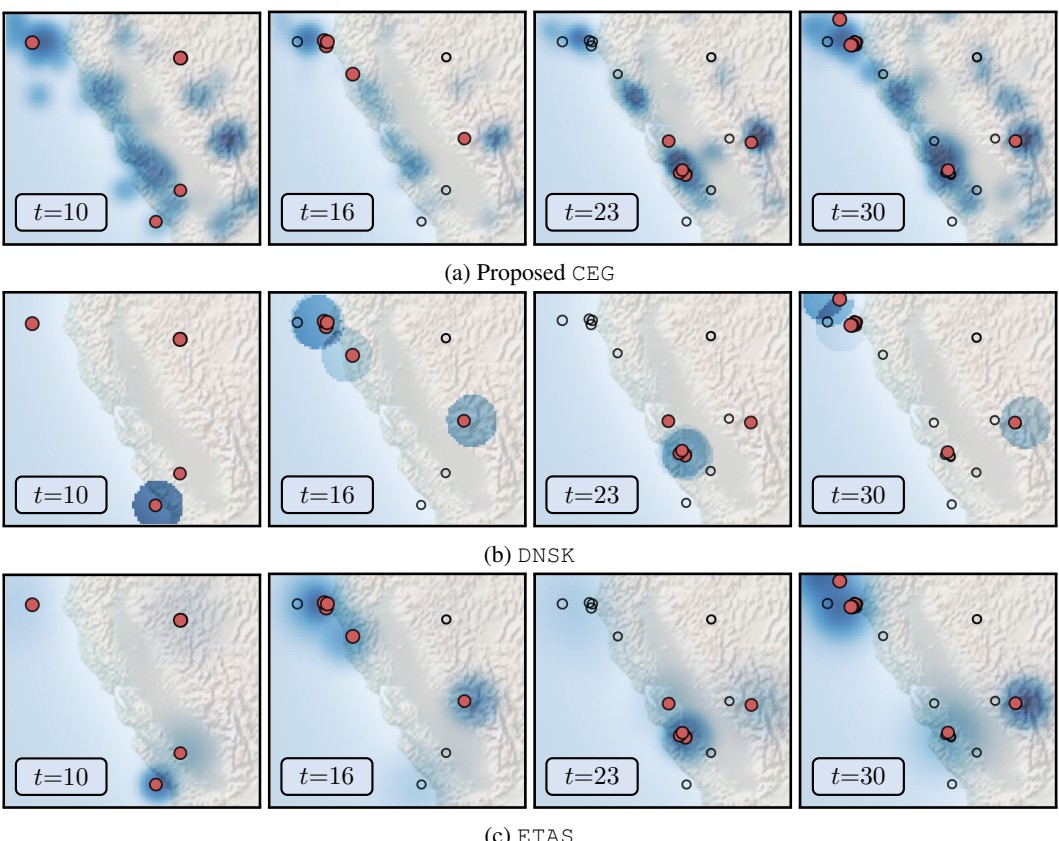

Figure E5: Estimated conditional PDFs of an actual earthquake sequence represented by shaded areas, with darker shades indicating higher conditional PDF values. Each row contains four sub-figures, arranged in chronological order from left to right, showing snapshots of the estimated conditional PDFs. The red dots represent newly observed events within the most recent time period, while the circles represent historical events.