# OpenReview forum: "Conditional Generative Modeling for High-dimensional Marked Temporal Point Processes"
_ICLR.cc/2024/Conference — Submitted to ICLR 2024_

### Official Review · Reviewer_LQDX · 2023-10-18

**Soundness:** 2 fair
**Presentation:** 3 good
**Contribution:** 2 fair
**Rating:** 3
**Confidence:** 3

**Summary:**

The paper proposes a new method to generate event sequences with high dimensional marks such as text and images. The method combines MTPP models with conditional density generator which directly models the generator function condition on history and a random vector from Normal distribution. This makes the approach more efficient (in terms of sample generation and prediction), and more expressive (by avoiding explicitly confining the conditional intensity function to a specific form).

**Strengths:**

Originality: The idea of making the mark to not restricted to discrete or continuous is somewhat novel. Some works have been done in spatio-temporal point process (i.e. Chan et al. NEURAL SPATIO-TEMPORAL POINT PROCESSES). It is interesting to see in high dimensional mark as text and image which are modern applications(i.e. Vision and NLP).  One paper in the domain is (Mehrasa A Variational Auto-Encoder Model for Stochastic Point Processes).  As far as I know,  this is the first work to do it with “GAN-type” generative models.

Quality: the overall okay. The authors attempt to include a comprehensive experiments from full synthetic, semi-synthetic and real applcaitions to support the main claim of the proposed model.

Clarity: Overall the presentation is fine. I understand the flow of the paper fairly well, from motivation, proposal of model, model itself, and experiments. I have questions about some details such as the comments from weakness.

significance. This can be an interesting idea – combined TPP with generated models, and benefit the TPP community.

**Weaknesses:**

Quality:
Methodology: the paper proposes a conditional generator for tpp with high dimension marks. Combining 1dimensional timestamp and high dimensional marks and learn to estimate the density (whether implicitly or explicitly) seems to be a little problematic. Error in estimating of timestamp will likely to be engulfed by error in estimating high dimension marks. (a similar example in causal inference for estimating outcome given (potentitally) high dimension covariates X and 1 D binary treatment {0,1}, i,e, g(y;X,T). )

Data and Analysis & Experimental Design: the author(s) should focus on experiments on high dimensional marks.
1.Experiments on synthetic data with 1D and 3D does not support the main claim. However it can be added to Appendix instead of main text.  1.Could the author generate tpp with high dimensional marks? (reasonably high like 50?)

2. For semi synthetic data, maybe the author(s) should try to find some quantitative metrics. I can “see” the performance of CEG is good on T-MNIST but   I cannot evaluate how good the generated series on T-CIFAR, even though it is better than DNSK.
3. Real data: a. Northern California earthquake catalog. It seems to it is spatiotemporal point process. Thus mark is 2-dimensional. B. Atlanta crime reports with textual description is an interesting application. Even so the mark is processed to be in 10 dimensional.
4. There is no evaluation of time in the above experiments.

Clarity: The key of the proposed model is equation (5).  However, to estimate the density conditional PDF in 5, it is unclear whether (6) or (7) are used in the experiment. Also in the experiment section only the density estimation of t without mark is shown. This is very limiting and does not support the proposed model.

**Questions:**

1.	See comments in Weakness.
2.	Datasets to validate. Could the authors find a dataset of sequences of video frames and perform experiments on such datasets so that the generated dynamics can be reflected?
3.	Baselines and ablation. Has the authors done experiments on removing timestamps so that x is just the mark from f(x|Ht(x)) in equation 5?

---

> ### Author Response · Authors · 2023-11-20
> **Response to reviewer LQDX (1/2)**
>
> We thank the reviewer for all the insightful comments, and we answer the questions on a point-by-point basis as follows:
>
> > Error in estimating of timestamp will likely to be engulfed by error in estimating high dimension marks.
>
> We thank and agree with the reviewer on the comment. When modeling events with high-dimensional marks, the variational learning strategy is adopted. Moreover, the reconstruction loss between the observed and generated events are measured across different domains (time, space, and mark, etc.), and these reconstruction losses are combined by a set of predefined weights, indicating the importance of different domains in specific problems. For example, if higher accuracy in time is preferred, a larger weight can be assigned to the time domain in the final loss function.
>
> > Could the author generate tpp with high dimensional marks? (reasonably high like 50?)
>
> We clarify that we do have the experiments of marked temporal processes with 784-dimension marks (T-MNIST) and 1024-dimensional marks (T-CIFAR). For example, each event in T-MNIST dataset is observed with a timestamp and an associated 784-dimension vector flattened from an image in the MNIST dataset. The model will treat each observed vector as a high-dimensional mark without knowing it as an image.
>
> We want to emphasize that the main novelty of our paper is that we are the very first paper that generalizes marked temporal point processes into high-dimensional mark space. While existing intensity-based methods suffer from the intractable integral in the model log-likelihood function when operating in high-dimensional data space, the proposed model-agnostic framework based on a generative architecture avoids computing the intractable integral during model estimation. This approach opens the door for effectively estimating high-dimensional marked point processes (we propose two learning strategies in the paper as examples, and more can be adopted), and is applicable in broad real-world scenarios with various problem settings or datasets.
>
> > For semi synthetic data, maybe the author(s) should try to find some quantitative metrics.
>
> We thank the reviewer for the comment. Traditionally, the evaluation of temporal events with high-dimensional marks presents its own challenges. For example, event generation with image or text is usually evaluated qualitatively by eyeballing the generated samples for their fidelity to real data and their ability to mimic sequence dynamics. In our study, we take the similar strategy and carefully design the semi-synthetic data and experiments, so that the quality of the generated samples can be effectively evaluated. For example, an effective model should not only capture the distribution of a single event mark (generating marks that resemble real images), but also capture the transition dynamics of event marks (mimic the Fibonacci sequence or the student's daily routine). Lastly, the superior quality of the generated samples by our model, in terms of the above two perspectives, has been demonstrated by the experimental results.
>
> We agree with the reviewer and acknowledge the importance of quantitative metrics in providing a more objective and comprehensive evaluation of our model's performance. To address this, we plan to introduce a novel metric, termed \textit{FID*}, inspired by the Frechet Inception Distance (FID score) commonly used in image generation studies [1] to evaluate the quality of generated images. This enhanced metric adapts the FID score to better suit our high-dimensional point process datasets by incorporating a temporal component, which is particularly important for point process data. This addition will enable a more rigorous comparison between our method and baseline models. Our revised manuscript will include the results obtained using this \textit{FID*} metric, aiming to offer a clearer, more quantitative evaluation of our model's performance on semi-synthetic datasets.
>
> ---
> [1] Martin Heusel, Hubert Ramsauer, Thomas Unterthiner, Bernhard Nessler, and Sepp Hochreiter. Gans trained by a two time-scale update rule converge to a local nash equilibrium. Advances in neural information processing systems, 30, 2017.

---

> ### Author Response · Authors · 2023-11-20
> **Response to reviewer LQDX (2/2)**
>
> > The key of the proposed model is equation (5). However, to estimate the density conditional PDF in 5, it is unclear whether (6) or (7) are used in the experiment. Also in the experiment section only the density estimation of t without mark is shown. This is very limiting and does not support the proposed model.
>
> We apologize for the confusion and would like to emphasize that the main novelty of the proposed framework is to approximate the log-likelihood function using generative models without computing the integral in Equation (3), which is notoriously difficult to evaluate and even intractable when the mark is high-dimensional. Our paper opens the door for effectively estimating high-dimensional marked point processes and is applicable in broad real-world scenarios. We clarify that Equations (6) and (7) are used in non-parametric learning and variational learning, respectively, which serve as two examples of estimating the high-dimensional marked point processes. More learning strategies, such as score-matching used in diffusion models, can also be adopted within our framework.
>
> We also highlight that the evaluated density functions of 3D synthetic and real earthquake datasets have been shown in Appendix E, demonstrating the proposed model's superior performance. Meanwhile, the density of data with high-dimensional marks is challenging to evaluate and even intractable.
>
> > Could the authors find a dataset of sequences of video frames and perform experiments on such datasets so that the generated dynamics can be reflected?
>
> We thank the reviewer for the suggestion. We want to point out that the sequences of video frames differ from the discrete event data we are dealing with. Specifically, there is no such definition of discrete and asynchronous events in video frames, which is the most unique trait of point process data. Meanwhile, the time intervals between events can be irregular and contain important knowledge about the underlying dynamics of the sequence. However, there exists a possibility of transforming video frames into point process data for specific problems. We can extract certain frames of the videos at different times if those frames contain key information about the video content. And each key frame can be defined as a discrete event, with associated image information as marks. We are open to the implementation on more interesting real-world applications in future research.
>
> > Has the authors done experiments on removing timestamps so that x is just the mark from $f(x|\mathcal{H}_{t(x)})$ in equation 5
>
> We clarify that the asynchronous timestamps for discrete events are the fundamental feature for marked temporal point process data, which cannot be removed in point process studies. For example, as a typical real-world application of point process modeling, the temporal effect of historical earthquake events plays a pivotal role in predicting the occurrences of future aftershocks and should be precisely captured. We highlight that point process modeling is used to analyze and model the occurrence of events over time, and most existing point process studies only focus on predicting future event times.

---

> > ### Comment · Reviewer_LQDX · 2023-11-20
> > **part 2/2 reply**
> >
> > For Q1, what i wanted clarification is what experimental results are from Equations (6) ? what are from  Equations (7)?
> > For Q2 & Q3, i understand difference between TPP and regularly space event sequences. I really hope to see some evidence which can validate the propose model. Thank you commenting on these two problems.

---

> > > ### Author Response · Authors · 2023-11-22
> > >
> > > >  what i wanted clarification is what experimental results are from Equations (6) ? what are from Equations (7)?
> > >
> > > The lower-dimensional results, such as temporal and spatio-temporal point processes (1D and 3D), in section 3.1 and earthquake in section 3.3 are obtained using Eq (6). The high-dimensional results in section 3.2 and text in section 3.3 are obtained using Eq. (7). We would like to emphasize that the proposed generative framework is completely general and model agnostic. It not only allows to model high-dimensional marked temporal point processes without estimating the integral in the log-likelihood, but also enables a more efficient simulation for high-dimensional marked events, which is computationally intractable by the traditional thinning algorithm.
> > >
> > > > I understand difference between TPP and regularly space event sequences. I really hope to see some evidence which can validate the propose model.
> > >
> > > As we indicated in both our response and our paper, there are two key criteria to access the effectiveness of our framework through the semi-synthetic experimental results: 1. The generated marks (images) have to "look real", indicating that the model is able to capture the marginal distribution of marks very well; 2. The transition between marks has to align correctly with our design (e.g., the third number equals to the sum of two preceding numbers), indicating that the model is capable of capturing the dependence on the history. Our experiments reveal that achieving even one of these criteria is challenging, with none of the current state-of-the-art methods meeting both requirements satisfactorily. Additionally, we agree that incorporating a quantitative metric like FID* would further validate our model's effectiveness.

---

> ### Comment · Reviewer_LQDX · 2023-11-20
> **part 1/2 reply**
>
> Thanks for your response.
>
> For Q 1, I am not sure what the  a set of predefined weights are. Maybe I missed from the paper. I am not yet convinced.
> For Q 2. how do you generate the timestamp and then an image?
> Thank you for address Q3 with a good potential metric.

---

> > ### Author Response · Authors · 2023-11-22
> >
> > Dear reviewer LQDX,
> >
> > We thank you for the comment. Please see our response below:
> >
> > >  I am not sure what the a set of predefined weights are.
> >
> > In our spatio-temporal experiments, we assign equal weight to each dimension; in the high-dimensional experiments, we assign higher weight (20) to the time and smaller weight to the marks (1) since the high-dimensional mark can dominate the loss easily as the reviewer pointed out in the review. Note that this is a user-defined variable and may vary from application to application and can also depend on the scale of data. We will include more detailed description about our experimental setup in the final version of our paper.
> >
> > > how do you generate the timestamp and then an image
> >
> > Our model generates time and marks (such as image) based on their observed history jointly. As we pointed out in our response to Reviewer GHUR. Our framework aim to capture a joint density distribution $p(t, m|\mathcal{H}_t)$, whereas the the majority of traditional point process models, such as RMTPP, divides the time and mark spaces. In other words, it models the product of two conditional distributions, $\widehat{p}(t|\mathcal{H}_t)$ and $\widehat{p}(m|\mathcal{H}_t)$. This separation can result in a more limited solution space and potentially less optimal outcomes.

---

### Official Review · Reviewer_AoCo · 2023-10-31

**Soundness:** 2 fair
**Presentation:** 3 good
**Contribution:** 3 good
**Rating:** 5
**Confidence:** 5

**Summary:**

This paper presents a model-agnostic generative framework to model event sequences with high-dimensional marks, a setting generally overlooked by prior works. Instead of directly defining a parametric form of the conditional intensity function (or conditional density function), they propose to estimate the distribution of events using samples generated from a conditional generator. The framework being introduced can be trained through different learning algorithms, and enjoys better computational efficiency compared to approaches relying on the thinning algorithm to simulate new events.

**Strengths:**

1) The proposed framework scales well to a setting with high-dimensional marks, an interesting research direction that has currently received little attention from the neural TPP community.
2) The framework being model-agnostic, it could be easily extended to other classes of models and learning algorithms than the ones presented in the paper.
3) The methodology proposed by the authors is presented clearly, which makes the paper intelligible and easy to follow.

**Weaknesses:**

1) In Section 2.2, you state "the proposed model enjoys considerable representational power, as it does not impose any restrictions on the parametric form of the conditional intensity $\lambda$ or PDF $f$." Several neural TPP models, such as FullyNN [], also do not impose any restrictions on the parametric form of the conditional intensity function. In what sense does your approach enjoy additional expressiveness compared to these models?

2) In section 2.3 regarding equation (5), you say "It is worth noting that this learning objective circumvents the need to compute the integral in the second term of (3)...". However, the negative log-likelihood expressed in terms of the events PDF still involves the conditional CDF evaluated at the observation window, which appears to have been omitted in this case.

3) For semi-synthetic and real world data, why don't you provide evaluation metrics to assess the performance of the different baselines as you did with the simulated datasets? Visualizations on hand-picked sequences are not enough to get a clear overview of the baselines performance.

4) From the training details provided in the Appendix, I understand that you fixed the hyper-parameters of the different models. Did you experiment with other hyper-parameters configurations as well ?

5) On Figure 6, you only show the results for the proposed CEG method, what about the other baselines (e.g. DNSK)?

**Questions:**

See the 'Weaknesses' section.

---

> ### Author Response · Authors · 2023-11-20
> **Response to reviewer AoCo (1/2)**
>
> We thank the reviewer for all the insightful comments, and we answer the questions on a point-by-point basis as follows:
>
> > In what sense does your approach enjoy additional expressiveness compared to these models?
>
> We thank the reviewer for the question. We point out that the majority of the existing literature tends to concentrate exclusively on one-dimensional, temporal-only point processes or adopt stringent parametric assumptions in modeling the event mark. For example, these methods often separate time and mark space, applying a specific parametric form to the influence kernel over mark space tailored to certain problems or datasets. This approach restricts the model's flexibility and wider applicability. Crucially, these methods struggle with generalization when it comes to data modeling in high-dimensional mark spaces due to the computational complexity.
>
> Take, for instance, the FullyNN method, which is suitable only for one-dimensional, temporal-only point processes. It necessitates the definition of the cumulative intensity function and the computation of its derivatives. This process may require additional assumptions about the function's smoothness and becomes challenging to generalize to multi-dimensional or high-dimensional mark spaces, especially due to the complexities involved in calculating the Jacobian matrix.
>
> On the contrary, our paper is the first study that generalizes marked temporal point processes into high-dimensional mark space, through a model-agnostic generative framework. The framework can be applied in modeling high-dimensional marks in broad real-world scenarios without any parametric restrictions. The inspiring empirical results in our experiments have also demonstrated the strong representative power of our model.
>
> > the negative log-likelihood expressed in terms of the events PDF still involves the conditional CDF evaluated at the observation window, which appears to have been omitted in this case.
>
> We thank the reviewer for the question. In our proposed framework, we do not calculate the integral over data space $\mathcal{X}$ in the second term of the original log-likelihood function shown in Equation (3), which is notoriously expensive to evaluate and even intractable when $\mathcal{X}$ is high-dimensional (as this is the main obstacle that hinders marked point process from being generalized in high-dimensional mark space). Instead, we approximate the $\log f_\theta$ in Equation (5) using generative models without calculating the integral. For example, the density function can be approximated using KDE in non-parametric learning (Equation 6 and Appendix B), or it can also be approximated by a surrogate objective known as evidence lower bound (ELBO) in variational learning (Equation 7 and Appendix C) using generated samples. Note that we can also take advantage of the cutting-edge techniques in other generative models to estimate the model log-likelihood, such as score-matching in diffusion models, as discussed in the last section.
>
> > Did you experiment with other hyper-parameters configurations as well?
>
> We thank the reviewer for the question and suggestion. As clarified in Appendix E, we take the best hyper-parameters indicated by their original papers for baselines considered in this paper. We are open to the idea of comparing more hyper-parameter settings if the reviewer thinks it is necessary for improving the paper's quality.
>
> > On Figure 6, you only show the results for the proposed CEG method, what about the other baselines (e.g. DNSK)?
>
> We thank the reviewer for the comment. We show Figure 6 as a case study to demonstrate our model's applicability in text data and also the capability to capture and simulate interpretable keywords for crime incidents with complex dynamics. Other baselines fail to model such complex event dependencies or generate meaningful crime patterns. Meanwhile, as we mentioned before, the assessment of the generated spatio-temporal-textual samples is heuristic and not well-defined. Thus there is no appropriate metric that can be applied here. Lastly, we can add more results using other methods if the reviewer thinks it is necessary for improving the paper's quality.

---

> ### Author Response · Authors · 2023-11-20
> **Response to reviewer AoCo (2/2)**
>
> > For semi-synthetic and real world data, why don't you provide evaluation metrics to assess the performance of the different baselines as you did with the simulated datasets? Visualizations on hand-picked sequences are not enough to get a clear overview of the baselines performance.
>
> We thank the reviewer for the comment. First, we want to clarify that the testing log-likelihood used in Table 1 is inapplicable in high-dimensional space for those baselines that rely on modeling the conditional intensity function, and we do not have the corresponding ground-truth to compute MRE of $f$ or $\lambda$.
>
> Traditionally, the evaluation of temporal events with high-dimensional marks presents its own challenges. For example, event generation with image or text is usually evaluated qualitatively by eyeballing the generated samples for their fidelity to real data and their ability to mimic sequence dynamics. In our study, we take the similar strategy and carefully design the semi-synthetic data and experiments, so that the quality of the generated samples can be effectively evaluated. For example, an effective model should not only capture the distribution of a single event mark (generating marks that resemble real images), but also capture the transition dynamics of event marks (mimic the Fibonacci sequence or the student's daily routine). Lastly, the superior quality of the generated samples by our model, in terms of the above two perspectives, has been demonstrated by the experimental results.
>
> We agree with the reviewer and acknowledge the importance of quantitative metrics in providing a more objective and comprehensive evaluation of our model's performance. To address this, we plan to introduce a novel metric, termed \textit{FID*}, inspired by the Frechet Inception Distance (FID score) used in image generation studies [1] to evaluate the quality of generated images. This enhanced metric adapts the FID score to better suit our high-dimensional point process datasets by incorporating a temporal component, which is particularly important for point process data. This addition will enable a more rigorous comparison between our method and baseline models. Our revised manuscript will include the results obtained using this \textit{FID*} metric, aiming to offer a clearer, more quantitative evaluation of our model's performance on semi-synthetic datasets.
>
> ---
> [1] Martin Heusel, Hubert Ramsauer, Thomas Unterthiner, Bernhard Nessler, and Sepp Hochreiter. Gans trained by a two time-scale update rule converge to a local nash equilibrium. Advances in neural information processing systems, 30, 2017.

---

### Official Review · Reviewer_GHUR · 2023-11-01

**Soundness:** 3 good
**Presentation:** 2 fair
**Contribution:** 2 fair
**Rating:** 3
**Confidence:** 3

**Summary:**

The proposed method is a generative model that outputs the time and the mark of the point process with a generator network, that is, maps noise to data. The learning is then performed using kernel density estimation or variation approximation. The method shows promising results on synthetic and real-world data.

**Strengths:**

The paper proposes a new conditional network to generate samples of a marked TPP directly. This is a nice alternative to the usual approaches of generating either time or mark first and conditioning the generation of the other. The generation is performed through a network that takes in the history embedding and a random noise vector that acts a source of noise similar to a generator in GANs. The training is not done in an adversarial way but rather using either kernel density estimation or variational inference. According to the empirical results this works surprisingly well on some synthetic data and a real-world scenario of earthquake prediction.

**Weaknesses:**

The model estimation seems like the most important contribution of the paper, although both rely on already known techniques. Equation 6 seems to include samples, something that is stated as a drawback for some of the competing methods. What is the computation cost of Algorithm 2, and compared to other methods?

In Appendix C it is stated that the second term of the ELBO is minimizing the NLL $p(x | z, h)$ which is the cross entropy between the observed and reconstructed event. But in this case, $x$ is both the mark and the time so I am not sure how this works out.

In 3.1 it says that only the conditional intensity can be computed in the baselines and the density has to be computed with numerical approximation. 1) How can your method compute the density in closed-form? 2) At least for fully NN model it is possible to compute the intensity and cumulative intensity in closed-form which will give you density.

As there is so much focus on thinning there seems to be a lack of baselines that do not require it, for example, having parametric density functions or (continuous) normalizing flows would allow immediate sampling. Using inverse CDF parameterization as well as diffusion models also allow direct sampling. The expressiveness of these models is still probably good enough for the tasks considered in the paper.

I am surprised with the results on synthetic data since at least fully NN model should be able to capture such simple intensity. Perhaps the model was not tuned correctly.

Section 3.2 setup is not really realistic. This is not a TPP problem but rather an image generation problem. Just because some methods cannot capture the complicated distribution *within* each mark does not mean that they cannot capture relations *between* marks -- what should have been tested. That is, the *temporal* point process here is easy and the conditional generation of marks is hard. I don't see why RMTPP augmented with a strong image generator could not beat proposed method. A more meaningful comparison would be some kind of dataset where each pixel in an image arrives at some time and they have some sort of interaction between each other, which is non-trivial.

The drawback of 3.2 is shown on in 3.3 since we have much simpler and lower dimensional real-world data.

The main issues are 1) the question of which part of the paper helps in getting the good results the most, is it avoiding learning using likelihood or the architecture? 2) Why some baselines are not performing so well on synthetic data and why stronger baselines are not used in 3.2? 3) Missing demonstration of generating proper high-dimensional marked point processes.

Minor:

- (Graves & Graves, 2012) citation is wrong, both the reference and the formatting of the reference "we opt for long short-term memory (LSTM) (Graves & Graves, 2012)".

**Questions:**

See weaknesses.

---

> ### Author Response · Authors · 2023-11-20
> **Response to reviewer GHUR (1/2)**
>
> We thank the reviewer for all the insightful comments, and we answer the questions on a point-by-point basis as follows:
>
> > The model estimation seems like the most important contribution of the paper, although both rely on already known techniques.
>
> We would like to clarify that model estimation is only part of the contribution of our paper. To the best of our knowledge, our paper is the first attempt that aims to generalize marked temporal point processes into high-dimensional mark space, and our main focus is to provide a model-agnostic approach through a novel and flexible generative framework without incurring expensive computational costs.
>
> The proposed learning algorithms serve as two examples of model estimation. To be specific, the non-parametric learning (Equation 6 and Appendix B) can be used to estimate model log-likelihood by directly approximating the density function through KDE in 1D and 3D data space, and variational learning (Equation 7 and Appendix C) approximates the log-likelihood by estimating a surrogate objective known as evidence lower bound (ELBO), which can be used in more broad scenarios with high-dimensional marks. We would also like to point out that alternative methods not covered by this paper, such as diffusion models with score-matching techniques, can also be applied within the proposed framework, as discussed in the last section.
>
> > Equation 6 seems to include samples, something that is stated as a drawback for some of the competing methods. Computation cost of Algorithm 2?
>
> We apologize for the confusion and would like to emphasize the main limitation of using Equation (3) is that the numerical integration for the integral of the intensity function ($\int_{\mathcal X}  \lambda(x| \mathcal{H}_{t(x)} ) dx$) is notoriously difficult to evaluate for point processes and even intractable when the data space $\mathcal{X}$ is high-dimensional. Hence this motivates us to explore new solutions that tackle the computational challenge using generative modeling. Here, KDE in Equation 6 only serves as an example, showing that the *log-likelihood can be estimated without computing the expensive integral in Equation (3).* Meanwhile, other learning methods, such as variational learning and the score-matching technique, can be adopted in this setting for efficient model estimation.
>
> Lastly, we agree with the reviewer that KDE can be expensive in a high-dimensional setting and thus may be limited in learning complex high-dimensional marked temporal point processes. To specify, given $N_T$ observed events, Algorithm 2 requires the complexity of $\mathcal{O}(L\cdot N_T)$ ($L$ is the sample size for estimating the density of each event using KDE), while existing estimation using Equation (3) require a complexity of $\mathcal{O}(N^d \cdot N_T)$ ($N^d$ is the number of sampled points for numerical integration). Our experiments only test the KDE methods on 1D and 3D synthetic datasets, and our results on high-dimensional datasets are obtained using variational learning. In the final version of the paper, we plan to move the KDE to the appendix to avoid further confusion.
>
> > How to compute the second term of the ELBO in Appendix C with $x$ as both the mark and time?
>
> In this study, we jointly model the event times and marks. The second term ''can be implemented as the reconstruction loss and calculated using generated samples'' both in time and mark space, as clarified in Appendix C. Specifically, the reconstruction loss between the observed and generated events are measured across different domains (time, space, and mark, etc.), and these reconstruction losses are combined by a set of predefined weights, indicating the importance of different domains in specific problems. For example, if higher accuracy in time is preferred, then a larger weight can be assigned to the time domain in the final loss function.
>
> > (1) How can your method compute the density in closed-form? (2) Fully NN it is possible to give you density in closed-form through the intensity and cumulative intensity.
>
> We thank the reviewer for the question and would clarify that we do not compute the density in closed-form. Instead, we approximate log-likelihood using generated samples. For example, the non-parametric learning (Equation 6, Appendix B) estimates model log-likelihood by directly approximating the density function through KDE, and variational learning (Equation 7, Appendix C) approximates the log-likelihood by estimating a surrogate objective known as evidence lower bound (ELBO).
>
> We highlight that the intensity and cumulative intensity in FullyNN can only be used in temporal-only point processes. Because it requires defining the cumulative intensity function and calculating derivatives, which can not be generalized to multi- or high-dimensional mark space. We also note that our proposed model provides a general and model-agnostic approach that extends marked point processes into high-dimensional space.

---

> ### Author Response · Authors · 2023-11-20
> **Response to reviewer GHUR (2/2)**
>
> > Lack of baselines that do not require thinning, for example, parametric density functions, (continuous) normalizing flows, inverse CDF parameterization as well as diffusion models.
>
> We thank the reviewer for the comments. With parametric density or conditional intensity function, the model expressiveness can be limited and thus lead to sub-optimal results, especially when complex event dependencies are involved. Our model has been demonstrated to outperform a parametric baseline $\texttt{ETAS}$ in our experiments.
>
> For normalizing flow or diffusion models, the computational cost will be extremely expensive if complicated model architectures are involved. Also, we would like to point out that there is a previous study [1] on normalizing flow in point processes, and their model still relies on the thinning algorithm for sampling.
>
> > Eq4: Section 3.2 setup is not really realistic. This is not a TPP problem but rather an image generation problem.  Why RMTPP augmented with a strong image generator could not beat the proposed method. A more meaningful comparison would be some kind of dataset where each pixel in an image arrives at some time and they have some sort of interaction between each other.
>
> We appreciate the reviewer's feedback and insights. It's important to note that our paper's experiments in both multi-dimensional and high-dimensional settings involve a simultaneous modeling of event dependencies across both time and mark space. This differs from a separate analysis of dependencies in time and mark space. Consequently, an RMTPP with an image generator may have limited effectiveness, as it separates time and mark space dependencies.
>
> To illustrate, consider a scenario where we're modeling a joint density distribution $p(t, m|\mathcal{H}_t)$. Our method aims to capture this joint distribution directly as $\widehat{p}(t, m|\mathcal{H}_t)$, whereas the RMTPP approach divides the time and mark spaces. It models the product of two conditional distributions, $\widehat{p}(t|\mathcal{H}_t)$ and $\widehat{p}(m|\mathcal{H}_t)$. This separation can result in a more limited solution space and potentially less optimal outcomes.
>
> We would also point out that the comparison example the reviewer suggested corresponds to a multivariate temporal point process with one-dimensional categorical marks, where each image pixel represents one event type, and events of different types come over a common timeline and interact with each other. While directly modeling a multivariate point process with a large number of event categories is expensive, our approach can still handle such a dataset flexibly and efficiently.
>
> > Minor:(Graves \& Graves, 2012) citation is wrong.
>
> We apologize for the confusion. We will correct the wrong formats and typos in the updated version of the paper.
>
> ---
> [1] Ricky TQ Chen, Brandon Amos, and Maximilian Nickel. Neural spatio-temporal point processes. arXiv preprint arXiv:2011.04583, 2020.

---

> > ### Comment · Reviewer_GHUR · 2023-11-22
> >
> > Thank you for your detailed answers.
> >
> > Since you are moving the KDE part to the appendix I will not comment on this further.
> >
> > The question on how authors compute ELBO is not answered. What are the specific losses, what are the predefined weights? What do you use in the earthquake example? You still need to evaluate the joint density somehow. So whatever you do, why not do it as $p(x | h)$ where $h$ represents the history? I understand why you use VI approach here but this might not be necessary. Additionally, this would not be the first VI model for point processes.
> >
> > The main disagreement here is on the task itself. High dimensional marks that you use in your synthetic experiments are an exercise in image generation, not TPPs. In my opinion, marks can be high dimensional like you use them but the challenge stems from their interactions, while here you don't have such a setting beyond a toy example. Spatio-temporal experiments are fine.
> >
> > I am keeping my score.

---

> ### Author Response · Authors · 2023-11-22
> **Reply to Reviewer GHUR's comment**
>
> We thank the reviewer for the comment.
>
> > How authors compute ELBO is not answered. What do you use in the earthquake example?
>
> The second term in ELBO is computed by the square of the L2 norm of the difference between the true sample $x$ and the reconstructed sample $\widehat{x}$, known as the reconstruction loss. Imagine we have observed events with 10-dimension marks; we denote the true sample as $x = (t, m_1, \dots, m_{10})$, where $t$ represents the event time and $m_j$ represents the $j$-th dimension of the event mark (Similar notation for $\widehat{x} = (\widehat{t}, \widehat{m} _1, \dots, \widehat{m} _{10})$). Thus the reconstruction loss is computed as $w_t(t-\widehat{t})^2 + w_m\sum _{j=1} ^{10} (m_j - \widehat{m} _j)^2$. Here, the $w_t$ and $w_m$ are the predefined weights indicating the importance of different domains in specific problems (*e.g.*, we set $w_t = 20$ and $w_m=1$ in semi-synthetic experiments to avoid the dominance of high-dimensional mark space in the loss function, as pointed out by Reviewer LQDX). The final reconstruction loss is summed over all the events. Thank you for the questions, and we will add the above clarification in the updated version of the paper.
>
> In earthquake example we use KDE to estimate the model log-likelihood instead of using variational learning.
>
> > why not do it as $p(x|h)$? I understand why you use VI approach here but this might not be necessary. Additionally, this would not be the first VI model for point processes.
>
> We would point out that directly estimating $p(x|h)$ is intractable in high-dimensional mark space. Specifically, Given the history $h$, what we will have, in the proposed generative framework, is those generated samples {$\widehat{x}$} drawn from the generator $g$. That means we do not have a model component for directly evaluating $p(x|h)$, and we can only estimate $p(x|h)$ using {$\widehat{x}$}. In the low-dimensional mark space, the $p(x|h)$ can be estimated by KDE with the generated samples, which is adopted for the earthquake data, while the KDE approach in high-dimensional mark space is computationally intractable for accurate density estimation. Meanwhile, variational learning serves as an effective and computationally efficient tool for approximating the density based on the generated samples, therefore we adopt it in the semi-synthetic datasets and the crime reports dataset.
>
> We do not claim we are the first VI model for point processes. As clarified in the response (1/2), we incorporate variational learning as an example for efficient and effective model estimation. Alternative methods not covered by this paper, such as diffusion models with score-matching techniques, can also be applied within the proposed framework.
>
> > High dimensional marks that you use in your synthetic experiments are an exercise in image generation, not TPPs. Marks can be high dimensional like you use them but the challenge stems from their interactions.
>
> We again apologize for the confusion, and we would like to point out that we are considering point process problems instead of image generation. The proposed model treats each observed data point as a high-dimensional mark (*e.g.*, a 784-dimension mark in T-MNIST data) without knowing it as an image. As we point out in our response to Reviewer AoCo (2/2) and to Reviewer LQDX (1/2) Question 3, the evaluation of temporal events with high-dimensional marks is usually evaluated qualitatively by eyeballing the generated samples for their fidelity to real data and their ability to mimic sequence dynamics. In our study, we take a similar strategy by designing the marks in semi-synthetic data as images, so that the quality of the generated samples can be effectively evaluated. For example, an effective model should not only generate marks that resemble real images, indicating that the model is able to capture the marginal distribution of marks very well; but also capture the transition dynamics of event marks (mimic the Fibonacci sequence or the student's daily routine).
>
> We also clarify that the marks in the semi-synthetic do have interactions with histories. For instance, in T-MNIST data, we design the number in the third image equals to the sum of numbers in the two preceding images; in T-CIFAR data, the distribution of future marks also follows a transition probability matrix given the history. It is not an exercise for image generation at given times, but a TPP problem for predicting future event times and marks. Our experiments reveal that achieving even one of these criteria is challenging, with none of the current state-of-the-art methods meeting both requirements satisfactorily. The superior quality of the generated samples by our model has been demonstrated by the experimental results.

---

> > ### Comment · Reviewer_GHUR · 2023-11-22
> >
> > Regarding the last point, I understand how you set up the experiment. The underlying process is some simple TPP for time and a simple interaction between marks (classes of images). The images themselves do not matter here since those are generated conditionally given the mark (of course, this is learned and not explicit). The TPP is simple, the final mark is complicated and the burden on the network is to learn to generate images. If I give you a more complicated dataset such as imagenet I assume this network would not produce high fidelity *marks*. My point being: the interactions between marks should be complicated and the number of marks should be large. What I refer to as marks can still be images or text or location but the interaction between different *types* of images should be more complicated. I can image that taking some pretrained image generation model and adding a simple TPP such that TPP models p(time, class) and feeds the generated class to conditionally generate an image would work well. This is why I believe this is more of an image generation problem. Perhaps I'm missing a bigger picture and you can provide a real-world example where this kind of model would be used.

---

> > > ### Author Response · Authors · 2023-11-23
> > > **Reply to Reviewer GHUR's comment**
> > >
> > > We thank the reviewer for reading our response and providing valuable feedback. We would clarify that there are few studies and datasets about high-dimensional marked point processes that have been scarcely explored. Therefore, we design these two semi-synthetic datasets with image marks that allow us to conduct experiments and effectively evaluate the performance of our model and baselines in high-dimensional mark space, with easy-to-understand data dynamics involved. Furthermore, to demonstrate the practical applicability of our model in capturing complex event dependencies, we also include the experiment on the crime report dataset. This case study serves as an example showing how our model can be applied in real-world scenarios and provides evidence of its effectiveness in modeling high-dimensional marks like text.
> > >
> > > We also acknowledge that in the future, our model can be adapted and experimented with whenever higher-quality real-world datasets become available, such as those containing temporal events/user activities with images and text from social media platforms. We are committed to incorporating more comprehensive datasets and experiments to enhance the quality of our research.
> > >
> > > Back to the modeling of low-dimensional marked events, our approach still enjoys superior performance that has never been achieved by other baselines. Take the earthquake dataset as an example. Our model captures the heterogeneous spatial-temporal event dynamics that align with the current understanding of the earthquake mechanics in Northern California, and is able to simulate event sequences that highly resemble the real earthquake activities, while other baselines fail to do so.

---

### Author Response · Authors · 2023-11-23
**Dear AC and reviewers**

We hope that our responses have appropriately addressed your concerns. Please let us know if you have any additional questions or comments. We would be more than happy to follow up with additional details. Thank you for dedicating your time to reviewing our efforts.

---

### Meta-Review · Area_Chair_L6yU · 2023-12-07

**Metareview:**

The authors propose using implicit generative models for marked point process data such as text and images. The reviewers raised many important concerns about the paper. One of the important ones raised by multiple reviewers is that the setup and experiments do not capture the intricacies and the complex interactions between marks, but it is reduced to a simpler conditional image generation problem. The novelty in this context was also found to be insufficient.

**Justification For Why Not Higher Score:**

The setup was found inadequate to capture the marked point process generation problem.

**Justification For Why Not Lower Score:**

N/A

---

### Decision · Program_Chairs · 2024-01-16

Reject